# Nanomaterials for the Removal of Heavy Metals from Wastewater

**DOI:** 10.3390/nano9030424

**Published:** 2019-03-12

**Authors:** Jinyue Yang, Baohong Hou, Jingkang Wang, Beiqian Tian, Jingtao Bi, Na Wang, Xin Li, Xin Huang

**Affiliations:** National Engineering Research Center of Industrial Crystallization Technology, School of Chemical Engineering and Technology, Tianjin University, Tianjin 300072, China; jyyang@tju.edu.cn (J.Y.); jkwang@tju.edu.cn (J.W.); beiqiantian@tju.edu.cn (B.T.); jingtaob@gmail.com (J.B.); wangna224@tju.edu.cn (N.W.); 2016207425@tju.edu.cn (X.L.)

**Keywords:** nanomaterials, heavy metal, wastewater, carbon-based nanomaterials, zero-valent metal, metal oxide, nanocomposite

## Abstract

Removal of contaminants in wastewater, such as heavy metals, has become a severe problem in the world. Numerous technologies have been developed to deal with this problem. As an emerging technology, nanotechnology has been gaining increasing interest and many nanomaterials have been developed to remove heavy metals from polluted water, due to their excellent features resulting from the nanometer effect. In this work, novel nanomaterials, including carbon-based nanomaterials, zero-valent metal, metal-oxide based nanomaterials, and nanocomposites, and their applications for the removal of heavy metal ions from wastewater were systematically reviewed. Their efficiency, limitations, and advantages were compared and discussed. Furthermore, the promising perspective of nanomaterials in environmental applications was also discussed and potential directions for future work were suggested.

## 1. Introduction

Water is one of the most important natural resources in the world, which is vital for the survival of all living beings and the development of humans. Along with the acceleration of industrialization and urbanization, the consumption of water is increasing rapidly and water scarcity problem has become an important constraint for economic development. In the meantime, water contamination, especially heavy metals pollution inside water, has become a global environmental issue. Heavy metals could be released into water mainly through the mining, electroplating, metallurgy, chemical plants, agriculture and household wastewater etc. Heavy metals such as Pb, Zn, Cu, Hg, etc. could pose a severe threat to human’s health because they can be accumulated biologically in the food chain [1]. For example, heavy metals could cause damages to the kidneys, mental and central nervous functions, lungs, and other organs [2,3,4]. Moreover, heavy metals can also exert adverse effects on the environment and other ecological receptors, as they cannot be degraded by microorganisms once they are released into the environment, on the contrary, they will accumulate through the food chain. Heavy metals are highly toxic [5], most of which are even reported to be carcinogenic [6]. Therefore, the removal of heavy metals from water is of great importance and has drawn tremendous attention. Up till now, numerous technologies have been developed to solve this problem, including chemical precipitation [7], ion exchange [8], adsorption [9], membrane filtration [10], electrochemical treatment [11], and so on. Besides, it is often the case that different techniques are combined for a better removal result [12,13]. Among the techniques discussed above, adsorption is one of the most extensively used techniques due to its low cost and simple operation. In recent years, porous metal-organic framework (MOF) materials have also shown their great superiority in eliminating hazardous substances from the environment due to their large/tunable porosity, pore functionality, and various pore structures etc. [14]. Ricco et al. has synthesized a magnetic nanocomposite based on aluminum MOF (MIL-53) and the composite exhibited excellent removal capacity towards Pb (II) with the value of 492.4 mg·g^−1^ [15]. Nowadays, nanomaterials have also provided a promising approach to removing heavy metals from wastewater.

Over the past decades, nanomaterials have gained a lot of attention. Numerous nanomaterials have been exploited in many fields, including electron devices [16], health care [17], energy [18], etc. The past decades have also witnessed the increasing applications of nanomaterials in the environmental protection field [19]. In general, nanomaterials are materials whose external dimensions are in the nanoscale (usually 1–100 nm) or those who have a nanoscale internal structure/surface [20]. Under the nanoscale, nanomaterials often exhibit some special properties, such as a surface effect, small size effect, quantum effect, and macro quantum tunnel effect [21]. These properties contribute to their extraordinary adsorption capacity and reactivity, both of which are favorable for the removal of heavy metal ions. So far, tremendous studies on nanomaterials have been carried out to investigate their applications on heavy metal water treatment and they have exhibited great potential as a promising alternative to adsorbing heavy metals from wastewater [22,23]. 

Based on the above background, this work reviews the latest development of nanomaterials which are used to remove heavy metals from wastewater. There have been some reviews regarding the water treatment of nanomaterials. For example, Marija et al. has summarized the graphene oxide nanocomposites for removing heavy metals [24]. Ihsanullah et al. has given a systematic review on the adsorption applications for heavy metals of carbon nanotubes [25]. Different parameters that could influence the adsorption behaviors have been discussed in this review. The current challenges and perspectives of carbon nanotubes were also well illustrated in this review. Hua et al. have summarized nanosized metal oxides for removing heavy metals [26]. However, these reviews were all focused on a single kind of nanomaterial, some of which were also outdated and missed the latest developments of nanomaterials for heavy metal treatment. In other reviews covering different kinds of nanomaterials, such as in the work of Wang et al., some nanomaterials for heavy metal removal including the carbon-based nanomaterials, metal/metal oxides nanoparticles, and polymer-supported adsorbents were simply reviewed [27]. These nanomaterials have shown their great potential for wastewater treatment due to their high adsorption capacity and selectivity, although this review was not that comprehensive. In Lee et al.’s review, metal oxide nanoparticles, carbon nanomaterials, and nanocomposites were discussed [28]. The preparation and characterization methods of these nanomaterials were introduced in this article. Still, the nanomaterials demonstrated in this review were incomprehensive, although this review has given a brief introduction on the measuring techniques of heavy metals. In this work, a systematic and comprehensive overview of the following nanomaterials: Carbon-based nanomaterials, zero-valent metal nanomaterials, metal oxide materials and nanocomposites, are presented. Some updated literatures related to the topic are especially presented here. The perspective of nanomaterials in heavy metal water treatment and the suggestion for future research direction are also discussed.

## 2. Adsorption Isotherms and Kinetics

In order to better illustrate the adsorption behaviors of the following nanomaterials, some models of adsorption isotherms and kinetics are briefly introduced in this chapter.

### 2.1. Adsorption Isotherms

Adsorption isotherms are very helpful for analyzing the adsorption capacities of the adsorbents. When the adsorption equilibrium is established, the relation between the amount of the adsorbates on the adsorbents and the equilibrium concentrations of the adsorbates under constant temperatures is called the adsorption isotherm [29]. There are various kinds of models for determining the adsorption isotherms, such as the Langmuir, Freundlich, Dubinin–Radushkevich, and Sips model etc. [30,31,32,33]. The Langmuir and Freundlich models are most extensively used and intensively discussed in this chapter.

#### 2.1.1. Langmuir Model

According to the Langmuir model, adsorption takes place uniformly on the active sites of the adsorbents, and once the adsorptive sites are occupied by the adsorbates, there will be no more adsorption behaviors on these sites [34]. The Langmuir model supposes that all the adsorption active sites have the same binding energy and each site is only able to bind a single adsorbate [35]. The linear form of the Langmuir model is expressed as [36]:(1)$$\frac{C_{e}}{q_{e}}=\frac{1}{bq_{m}}+\frac{C_{e}}{q_{m}}$$
where *q_e_* is the equilibrium adsorption capacity of the adsorbent (mg·g^−1^), *C_e_* is the equilibrium concentration of the adsorbate (mg·L^−1^), *q_m_* is the saturated single layer adsorption capacity (mg·g^−1^), and *b* is the adsorption equilibrium constant.

#### 2.1.2. Freundlich Model

The Freundlich isotherm model is another empirical equation which can be used to describe the non-ideal sorption behaviors [37]. The Freundlich model was proven to be consistent with the exponential distribution of active centers, characteristic of heterogeneous surfaces [34]. Unlike the Langmuir model, the Freundlich model is based on the multilayer adsorption and its linear form can be expressed as [38]:(2)$$\ln q_{e}=\left(\frac{1}{n}\right)\ln C_{e}+\ln K_{F}$$
where *q_e_* is the equilibrium adsorption capacity of the adsorbent (mg·g^−1^), *C_e_* is the equilibrium concentration of the adsorbate (mg·L^−1^), *K_F_* is the Freundlich constant (index of adsorption capacity), and *n* is also a Freundlich constant (index of adsorption intensity or surface heterogeneity).

#### 2.1.3. Sips Model

The Sips isotherm is a hybrid model of the Langmuir and the Freundlich isotherms [33]. The Sips model is used to predict the heterogeneous adsorption systems and can avoid the limitation of the rising adsorbate concentration associated with the Freundlich isotherm model [39,40]. At low adsorbate concentrations, the Sips isotherm effectively reduces to the Freundlich isotherm. While at high adsorbate concentrations, the Sips model predicts a monolayer sorption capacity characteristic of the Langmuir isotherm [41]. The Sips model could be expressed as [42]:(3)$$q_{e}=\frac{q_{m}{\left(K_{S}C_{e}\right)}^{n_{S}}}{1+{\left(K_{S}C_{e}\right)}^{n_{S}}}$$
where *q_e_* is the equilibrium adsorption capacity of the adsorbent (mg·g^−1^), *C_e_* is the equilibrium concentration of the adsorbate (mg·L^−1^), *q_m_* is the Sips maximum adsorption capacity (mg·g^−1^), *K_S_* (L·mg^−1^) is the Langmuir equilibrium constant and *n_S_* is comparable to the Freundlich heterogeneity factor *n_F_* (*n_S_* = 1/*n_F_*).

### 2.2. Adsorption Kinetics

The determination of kinetics is vital for the design of adsorption systems and the reaction rate controlling step as the chemical reaction occurs [43]. Adsorption kinetics describe the relations between the amount of adsorbates adsorbed on the adsorbents (*q_t_*) and the contact time (*t*). The commonly used adsorption kinetics include the pseudo-first-order and pseudo-second-order kinetic model. The pseudo-first order model describes the adsorption of liquid–solid phase systems based on the adsorption capacity [44], while the pseudo-second order model is based on the adsorption capacity of the solid phases.

For the pseudo-first order model of Lagrange, it can be generally expressed as follows [45]:(4)$$\frac{dq_{t}}{dt}=K(q_{e}-q_{t})$$
The integral form of this equation is given as follows:(5)$$\log(q_{e}-q_{t})=\log q_{e}-\frac{k_{1}}{2.303}t$$
where *q_e_* is the equilibrium adsorption capacity of the adsorbent (mg·g^−1^), *q_t_* is the adsorption capacity (mg·g^−1^) when the contact time is *t*, *K* and *k*_1_ are the rate constants of the pseudo first-order adsorption model.

The pseudo-second-order model can be expressed as follows [43]:(6)$$\frac{dq_{t}}{dt}=k_{2}{\left(q_{e}-q_{t}\right)}^{2}$$

The integral form of this equation is given as follows:(7)$$\frac{t}{q_{t}}=\frac{1}{k_{2}q_{e}{}^{2}}+\frac{1}{q_{e}}t$$
where *q_e_* is the equilibrium adsorption capacity of the adsorbent (mg·g^−1^), *q_t_* is the adsorption capacity (mg·g^−1^) when the contact time is *t*, *k*_2_ is the rate constant of the pseudo second-order adsorption model.

## 3. Nanomaterials for Removing Heavy Metals

### 3.1. Carbon-Based Nanomaterials

Carbon-based nanomaterials were initially applied in the electronics industry owing to their extraordinary thermal and electrical properties [46]. Nevertheless, some other exceptional properties they exhibited, such as a large surface area, ease of chemical or physical modification, ability of removing both organic, and inorganic pollutants have made carbon-based nanomaterials potential alternatives for treating wastewater [47]. Here, two major carbon-based nanomaterials are mainly presented—carbon nanotubes-based and graphene-based nanomaterials. 

#### 3.1.1. Carbon Nanotubes

Carbon nanotubes (CNTs) have been extensively investigated during the last decades and they were reported to exhibit many exceptional properties, including optical, electronic, vibrational, mechanical and thermal properties [48]. There have been numerous reports on their applications for the removal of heavy metals from wastewater [49]. Carbon nanotubes, basically divided into single-walled CNTs (SWCNTs) and multi-walled CNTs (MWCNTs) [50], are one kind of carbon-based materials whose lengths are about hundreds to thousands of nanometers and diameters are about 1–3 nm [51]. Carbon nanotubes have many superiorities in treating heavy metal wastewater, mainly including the large specific surface area, high adsorption capacity, and fast adsorption kinetics [52]. Carbon nanotubes were reported to have excellent adsorption effects towards Mn (VII), Tl (I), Cu (II), Pb (II), Cr (VI), etc. [53,54,55,56,57]. The possible adsorption active sites of carbon nanotubes are mainly comprised of outside surface, interstitial channels, internal sites, and external groove sites. Figure 1 has given an illustration of adsorption sites based on the adsorption of single-walled carbon nanotubes (SWNTs) bundles [58].

To improve the adsorption capacity of CNTs towards heavy metals, functional groups, such as –COOH, –NH_2_, –OH, etc. are commonly introduced onto the surface of CNTs by means of chemical modification, heat treatment or endohedral filling [59]. For example, it was reported that oxidants such as KMnO_4_, HNO_3_, H_2_SO_4_, and NaOCl could improve the adsorption capacity remarkably by modifying the surface of CNTs [60]. Mohamed et al. reported the removal of Hg (II) by employing a functionalized-CNTs absorbent [61]. Allyl triphenyl phosphonium bromide and glycerol, which were used to form deep eutectic solvent (DET), were sonicated with pre-oxidized CNTs to prepare the new functionalized CNTs. The batch adsorption experiment results showed that the optimal condition was pH = 5.5 and the contact time was 28 min. The corresponding maximum adsorption capacity for Hg (II) was determined to be 186.97 mg·g^−1^ by using the Freundlich isotherm model and the adsorption process followed the pseudo-second-order kinetics. Xu et al. have given a thorough review on functionalized carbon nanotubes for adsorbing heavy metals from wastewater, covering the preparation, application, and mechanism of modified CNTs [62]. Carbon nanotubes could also be combined with other supports to make better nanocomposites. For example, Zhan et al. prepared a novel magnetic amino-functionalized Fe_3_O_4_/carboxylic multi-walled CNTs hybrid by using a one-pot solvothermal method [63]. The novel CNTs-based nanocomposite showed an extremely high separation efficiency towards Cu (II) in batch adsorption tests, which resulted from the synergistic effect between the CNTs and the amino groups. The maximum adsorption capacity towards Cu (II) was calculated to be 30.49 mg·g^−1^ according to the Langmuir model. Moreover, the adsorbent could be separated from the wastewater with the aid of an external magnetic field. 

Although using CNTs to remove heavy metals from wastewater has many advantages, it still has a few drawbacks in many aspects. Firstly, the high costs of CNTs hinder their commercial use. A lot of work needs to be done to develop both effective and cost-saving CNTs. Moreover, it is usually difficult to separate CNTs from wastewater after the adsorption and this would increase the treatment costs and the risk of secondary pollution. Finally, the toxicological study of CNTs is also in high demand [59].

#### 3.1.2. Graphene Nanomaterials

Graphene, as the first 2D atomic crystal available to us, is another important carbon-based nanomaterial which can also be used to remove heavy metals from wastewater. It owns many extraordinary properties, such as mechanical strength, stiffness and elasticity, electrical and thermal conductivity and so on [64], contributing to its extensive applications in many aspects. Besides, graphene oxide (GO) and reduced graphene oxide (RGO) are two kinds of graphene-based nanomaterials which can also be used to remove heavy metals from wastewater. GO is the oxidation product of graphene and it contains miscellaneous oxygen-containing functional groups, such as hydroxyl, carboxyl, epoxide and carbonyl functional groups, which make it possible to remove heavy metals [65]. RGO, the reduction product of GO, commonly has more defects than the pristine graphene and is more easily to be modified by functional groups, such as –OH, –COOH, etc. [66]. Structure diagrams of some graphene-derived materials are given in Figure 2 [67]. The mechanism for these graphene-based nanomaterials to remove heavy metals lies in their large specific surface areas and some other extraordinary properties, such as ample functional groups, e.g., –CH(O)CH–, –OH, and –COOH etc., high negative charge density and highly hydrophilic characteristics [68]. 

Graphene and GO/RGO-based materials have been extensively reported to remove heavy metals from wastewater. For example, Wang et al. studied different factors on the adsorption performance of GO for the removal of heavy metals in batch tests, including pH, the dosage of the adsorbent, contact time, temperature, and coexisting ions [69]. The results showed that the adsorption process fitted the Langmuir isotherm and pseudo-second-order kinetic model well and the maximum adsorption capacity of Zn (II) was up to 246 mg·g^−1^, indicating that GO was an effective adsorbent for Zn (II). Zhao et al. synthesized few-layered graphene oxide nanosheets by using a modified Hummers method and used them to adsorb Cd (II) and Co (II) in water by the batch method [70]. It indicated that the adsorption effect was strongly influenced by pH and the presence of humic acid in an aqueous solution could reduce the adsorption of Cd (II) and Co (II). The maximum adsorption capacities of Cd (II) and Co (II) onto GO were 106.3 and 68.2 mg·g^−1^ respectively. The thermodynamic parameters of this adsorption process were also calculated and the results suggested that Cd (II) and Co (II) adsorptions on GO nanosheets were endothermic and spontaneous.

Moreover, there is an increasing number of reports on the graphene-based nanocomposites which are used to remove heavy metals from water. Some of the graphene-based nanocomposites are summarized in Table 1. As GO is well dispersed in water, it is difficult to separate it from the aqueous solution. Recently, Arshad et al. synthesized a novel graphene-modified absorbent which can solve this problem [71]. First, calcium alginate (CA) beads were embedded into the graphene oxide and then were further reduced by polyethylenimine to increase the adsorption capacity towards heavy metals. The adsorption experiment was carried out batch-wise in a shaker bath. The highest adsorption capacities of this GO-based nanocomposite towards Pb (II), Hg (II), and Cd (II) were 602, 374, 181 mg·g^−1^ respectively according to the Langmuir isotherm, indicating a high superiority material for removing these three ions. The adsorption kinetics were found to follow the pseudo-second-order kinetics and the adsorption thermodynamic parameters indicated that the adsorption process could be attributed to physicochemical adsorption. The functionalized beads reduced by polyethylenimine exhibited a higher adsorption ability compared with the non-functionalized beads as a result of synergetic effect. Moreover, the reusability of the adsorbent was also studied and the results showed that the removal efficiency for Pb (II) remained at 75–80% even after five cycles. Vilela et al. built novel graphene oxide-based microbots (GOx-microbots) which could serve as self-propelled systems for capturing, transferring, and removing heavy metals [72]. GO, Ni, and Pt composed the structure of microbots. The results indicated that the mobile GOx-microbots could remove Pb (II) ten times more efficiently than that of nonmobile GOx-microbots and the concentration of Pb (II) could be reduced from 1000 ppb down to below 50 ppb in 60 min after the treatment. The GOx-microbots could also be reused after removing the lead from the surface of the microbots.

However, most studies on graphene-based nanomaterials are at a preliminary stage of research, and research on the practical application of these materials in industrial wastewater treatment are still lacking, especially for actual wastewater containing multiple pollutants. Besides, the recycle and reuse of graphene-based nanomaterials also demand further investigation from an economic perspective [68].

### 3.2. Silica-Based Nanomaterials

Silica-based nanomaterials are another kind of important nanomaterial for removing heavy metals due to their properties, such as non-toxicity and excellent surface characteristics [85]. Nanosilica can be surface modified by groups like –NH_2_, –SH, etc., or serve as the support of nanocomposites. For example, Kotsyuda et al. synthesized silica nanospheres which were biofunctionalized by 3-aminopropyl and phenyl groups and investigated their removal effects towards Cu (II) and cationic thiazine dye in static mode [86]. The results indicated that the functionalized silica nanospheres possessed enhanced adsorption capacities towards Cu (II) and methylene blue compared with the amino functionalized nanosilica. The adsorption capacity of biofunctionalized nanosilica towards methylene blue was almost twice the value of similar amino silica nanoparticles. This modified nanomaterial also exhibited a decent antibacterial activity. Najafi et al. has investigated the removal effects for Cd (II), Ni (II), and Pb (II) in a batch mode by employing three silica-based nanomaterials, including the amino functionalized silica gel (NH_2_-SG), amino functionalized silica nano hollow sphere (NH_2_-SNHS) and non-functionalized silica nano hollow sphere (SNHS) [87]. The result indicated that the adsorption capacities of these three nanomaterials followed the order: NH_2_-SNHS > NH_2_-SG > SNHS and the trend of metal adsorption was Pb (II) > Cd (II) > Ni (II). The maximum adsorption capacities for Pb (II), Cd (II), and Ni (II) by using NH_2_–SNHS were 96.79, 40.73, and 31.29 mg·g^−1^ respectively. The adsorption isotherms were found to be correlated with the Langmuir-Freundlich (Sips) isotherm well and the kinetic data fitted the pseudo-second-order well. Besides the surface modification, silica has also been extensively reported to prepare the nanocomposites, among which the magnetic silica materials have received a lot of attention. Pogorilyi et al. has successfully coated the magnetite particles with silica layers by means of the Stöber reaction and showed its application potential in a broad industrial scale [88]. Some of the magnetic silica materials will be discussed in a later chapter. In a study carried out by Mahmoud et al., nanopolyaniline and crosslinked nanopolyaniline were immobilized onto the nanosilica to make nanocomposites, Sil-Phy-NPANI and Sil-Phy-CrossNPANI [85]. The adsorption effects of Sil-Phy-NPANI and Sil-Phy-CrossNPANI towards Cu (II), Cd (II), Hg (II), and Pb (II) were compared by using a batch technique. The highest adsorption capacities of Sil-Phy-NPANI for Cu (II), Cd (II), Hg (II), and Pb (II) were 1700, 800, 600, and 900 μmol·g^−1^ respectively, while the adsorption capacities of Sil-Phy-CrossNPANI for these four ions were 1650, 1050, 1350, and 1450 μmol·g^−1^ respectively by calculation from Langmuir isotherm. The results indicated that Sil-Phy-CrossNPANI could serve as an efficient adsorbent for Cd (II), Hg (II), and Pb (II).

### 3.3. Zero-Valent Metal-Based Nanomaterials

Zero-valent metal nanoparticles have exhibited their potential in water treatment and remediation in recent years. For example, Ag nanoparticles have been used to disinfect wastewater due to their antimicrobial ability [89]. Nanosized zero-valent zinc was reported to have an excellent degradation ability towards dioxins [90]. As for the heavy metal ions treatment, zero-valent iron was most extensively investigated and mainly discussed in this part. Moreover, some other nanosized noble metals were also discussed here.

#### 3.3.1. Zero-Valent Iron

Nanoscale zero valent iron (nZVI) is a composite consisting of Fe (0) and ferric oxide coating (Figure 3) [91]. It has received increased attention as a novel adsorbent to treat various kinds of heavy metals, such as Hg (II), Cr (VI), Cu (II), Ni (II), Cd (II), etc. since it came out [92,93,94]. Basically, Fe (0) provides the reducing ability while the ferric oxide shell offers the sites of reactive and electrostatic interaction with heavy metals. Besides, the particle size of nZVI is controllable and there are abundant reactive sites on the surface [95]. The high reducing capacity and large specific surface area contribute to the superior performance of nZVI in removing heavy metals from wastewater [96]. 

It is worth noting that the removal mechanism of nZVI for different heavy metals could vary according to the standard potential E^0^ of the heavy metals [96]. For example, for Pb (II), whose E^0^ is slightly more positive than that of Fe (II), the removal mechanism was mainly comprised of reduction and sorption, which can be expressed as follows [97]:(8)$$\mathrm{Reduction}:\equiv {\mathrm{Fe}}^{0}{+M}^{2+}\to \equiv {\mathrm{Fe}}^{2+}{+M}^{0}$$
(9)$$\mathrm{Sorption}:\equiv {\mathrm{FeOOH}+M}^{2+}\to \equiv {\mathrm{FeOOM}}^{+}{+H}^{+}$$

While for heavy metal ions such as Cr (VI) whose E^0^ is much higher than that of Fe (II), the dominated removal mechanisms were reduction and precipitation. The standard potential E^0^ of some environmentally relevant metals are given in Table 2 [98].

Although using nZVI to remove heavy metal has many advantages, its shortcomings cannot be neglected. NZVI was reported to be oxidized with oxygen and water in an aqueous solution, slowing down or hindering the reduction process of the heavy metals [99]. nZVI was also reported to aggregate easily, resulting in the decrease of the reaction surface area and mobility [91]. Besides, the separation of nZVI from wastewater is difficult. In order to improve the performances of the nZVI, various kinds of modification strategies have been developed, such as surface chemical modification or doping nZVI with other metals (Pd, Cu, Ni, Pt, etc.) [100]. For example, Huang et al. synthesized a novel nZVI-modified material by combing nZVI with sodium dodecyl sulfate (SDS) which is one kind of anionic surfactant and possesses excellent abilities of migration and dispersion [101]. The maximum removal capacity of this novel nZVI material towards Cr (VI) was 253.68 mg·g^−1^ in a batch adsorption experiment, indicating a promising adsorbent with an improved adsorption capacity and a decreased aggregation. The adsorption process was found to obey the Freundlich model and pseudo-second-order kinetic model well. Different factors such as pH, contact time, dosage, and initial concentration were also investigated and a maximum removal efficiency of 98.919% could be achieved under optimum conditions. Su et al. investigated the removal of both Cd (II) and nitrate in water in a batch mode by using nZVI and Au-doped nZVI nanoparticles [102]. By using Au-doped nZVI, the nitrite yield ration reduced from nitrate could be decreased significantly compared with the bare nZVI while the removal ration of Cd (II) remained at a high level. This result indicated that the Au-doped nZVI could be employed to treat wastewater containing both Cd (II) and nitrate. In addition to the two modification methods discussed above, nZVI-based nanocomposites are also receiving increasing attention. For example, Zarime et al. prepared a new nZVI-based nanocomposite by using low-cost bentonite to treat Pb (II), Cu (II), Cd (II), Co (II), Ni (II), and Zn (II) in water [103]. The introduction of bentonite to nZVI could decrease the aggregation of nZVI particles and provide the nZVI particles more adsorbing sites for heavy metals. Bentonite-nZVI composite exhibited a higher removal capacity towards these heavy metals compared with the mere bentonite. Not only employed in the laboratory level, there were also reports on the field tests of nZVI to remedy the groundwater in situ (Figure 4) [104].

#### 3.3.2. Ag Nanoparticles

Unlike nZVI, reports on other metallic nanoparticles which could be used to remove heavy metals are not sufficient. There have been several reports about the interaction between Ag nanoparticles and Hg (II) [105,106]. Although the reactivity between Hg (II) and the bulk silver is not high, Ag nanoparticles can exhibit a higher reactivity because the reduction potential of Ag decreased with the diminution of the particle size [107]. E. Sumesh et al. synthesized a novel silver nanoparticle-based adsorbent by coordinating Ag with mercaptosuccinic acid (MSA) [108]. Two different materials were prepared and investigated by varying the ratio of Ag to MSA. The results demonstrated that 1:6 Ag@MSA had a higher removal capacity towards Hg (II) (800 mg·g^−1^) compared with the common adsorbents. Furthermore, the authors stated that the cost for removing Hg (II) by employing Ag@MSA was competitive, indicating that Ag@MSA could be used as a promising alternative in the removal of Hg (II). 

#### 3.3.3. Au Nanoparticles

The affinity of Hg towards Au was generally recognized due to the fact that they could form AuHg, AuHg_3_, and Au_3_Hg [109]. Lisha et al. investigated the removal effect towards Hg (II) by using gold nanoparticles which were supported on aluminum [110]. Both batch and column tests were carried out in this work. NaBH_4_ was used to reduce Hg (II) to Hg (0) and the results showed that the removal capacity of Au nanoparticles towards Hg (0) reached up to 4.065 g·g^−1^, which was much higher than the ordinary adsorbents. The expense of using this kind of Au nanoparticles to treat Hg (II) was estimated to be low and the used Au nanoparticles could be recovered efficiently, demonstrating that Au nanoparticles supported on aluminum could be applied to practical wastewater treatment. In another study, Jiménez et al. developed citrate-coated Au nanoparticles which could be used to treat Hg (II) in water (Figure 5) [111]. The citrate ions served as a weak reducing agent which could reduce Hg (II) to Hg (0), thus evading the employment of NaBH_4_. The removal experiments showed that the concentration of Hg (II) could be decreased from 65 ppb to 1–5 ppb. The final product after the removal was Au_3_Hg alloy which could be treated at a high temperature or pressure later to recover Au [112].

### 3.4. Metal Oxide-Based Nanomaterials

Nanosized metal oxides possess many exceptional properties, such as a high removal capacity and selectivity towards heavy metals. Thus, they have great potential as promising adsorbents for heavy metals. Generally speaking, metal oxides-based nanomaterials include nanosized iron oxides, manganese oxides, zinc oxides, titanium oxides, aluminum oxides, magnesium oxides, cerium oxides, and zirconium oxides, etc.

#### 3.4.1. Iron Oxides-Based Nanomaterials

Iron oxide-based nanomaterials have received increasing attention in removing heavy metals from wastewater these years [113]. Iron is the fourth most abundant element in the Earth’s crust [114]. The abundance of the iron element and the simplicity of synthesizing iron oxides contribute to the wide investigations on iron oxide-based nanomaterials. The most investigated iron oxides are goethite (α-FeOOH), hematite (α-Fe_2_O_3_), maghemite (γ-Fe_2_O_3_), magnetite (Fe_3_O_4_), and hydrous iron oxides (HFO) [26], which will be discussed below. For α-FeOOH, α-Fe_2_O_3_, γ-Fe_2_O_3_, and HFO, the valences of iron are all trivalent, while for Fe_3_O_4_, iron exists both in bivalence and trivalence. 

##### Goethite (α-FeOOH)

Goethite (α-FeOOH), which already exists as a mineral in nature, is proven to be a competitive adsorbent towards heavy metals, thanks to its high adsorption efficiency, environmental safety and low cost [115]. Sun et al. prepared nanoscale α-FeOOH by using different ferrous and ferric salts and employed it to remove uranium in water [116]. The results indicated that nanoscale α-FeOOH had a much higher removal capacity towards uranium than that of non-nanoscale α-FeOOH, especially in the pH ranging from 5.5 to 7.5. Chen et al. synthesized nanoscale goethite which could serve as both a photocatalyst and adsorbent towards heavy metals by using a coprecipitation method [117]. The batch adsorption experiment results showed that the nanoscale goethite exhibited some photocatalyst activity towards methylene blue solution under the irradiation of UV-light. Meanwhile, this newly-prepared nanoscale goethite also possessed a high adsorption capacity towards Cu (II) (149.25 mg·g^−1^). The adsorption data fitted the pseudo-second-order equation and the Langmuir isotherm well. Thermodynamic data indicated that the adsorption process was spontaneous. Khezami et al. synthesized goethite nanocrystalline powders by using the high-energy ball milling method and employed them to treat Cd (II) in batch adsorption experiments [118]. The adsorption effect was greatly influenced by several adsorption factors and the maximum adsorption capacity of 167 mg·g^−1^ could be obtained when pH = 7 and the temperature was 328 K (55 °C). The adsorption data were in good agreement with both the Langmuir and Freundlich isotherms and the adsorption kinetics obeyed the pseudo-second-order model. The thermodynamics were also estimated and the results indicated that the adsorption process was spontaneous and endothermic. Goethite has also been reported to remove other heavy metals such as V (V), Mn (II), Ni (II), Co (II), Zn (II), Th (II) etc. [119,120,121].

##### Hematite (α-Fe_2_O_3_)

Hematite (α-Fe_2_O_3_) is the most stable iron oxide and is highly resistant to corrosion [122]. Hematite nanoparticles have shown their application potentials in many fields, such as lithium ion batteries [123], environmental indicators [124], and catalysis [125]. What is more, hematite nanoparticles are also proven to be effective adsorbents towards heavy metals [126,127,128]. Adegoke et al. investigated the effect of morphologies of hematite nanoparticles on the removal of Cr (VI) [127]. Different morphologies of nano-hematite, including hexagonal, plate-like, nano-cubes, sub-rounded and spherical were synthesized and the removal capacities towards Cr (VI) were in the range of 6.33–200 mg·g^−1^. The results indicated that morphologies play a significant role in the adsorption capacities of hematite. Shipley et al. studied the adsorption capacities of nano-hematite towards Pb (II), Cd (II), Cu (II), and Zn (II) [129]. Different operating factors, including the dosage of adsorbents, temperature, and multiple metal species were investigated. The affinity between the heavy metals and the adsorbent obeyed the following order: Pb (II) > Zn (II) > Cd (II) > Cu (II). The adsorption data fitted the pseudo-second-order rate model well, which indicates that the adsorption rate on the surface of the adsorbent is the rate-determining step. The thermodynamic data demonstrated that the adsorption for Pb (II), Cd (II), and Cu (II) was endothermic while that for Zn (II) was exothermic. The results showed that nano-hematite was an effective adsorbent which could remove multiple heavy metals from water simultaneously. Recently, superparamagnetic hematite nanoparticles were synthesized and used to treat the acid mine drainage (AMD) containing Al (III), Mg (II), Mn (II), Zn (II), Ni (II) etc. in a batch mode [130]. The results showed that nano-hematite could totally remove Al (III), Mg (II), and Mn (II), and could remove over 80% of Ni (II) and Zn (II). Given that nano-hematite possesses many merits such as non-toxicity, high stability, and an excellent metal adsorption capacity, it is a promising adsorbent to treat wastewater containing heavy metals.

##### Maghemite (γ-Fe_2_O_3_)

Maghemite (γ-Fe_2_O_3_) nanoparticles have been reported extensively to treat heavy metals in wastewater [131,132,133]. The advantages of maghemite nanoparticles for treating heavy metals lie in many aspects. Maghemite nanoparticles possess a large surface area which contributes to their high adsorption capacity. Besides, maghemite nanoparticles can be separated from wastewater easily after treatment by adding a magnetic field. Moreover, the synthesis of maghemite nanoparticles is simple and they are environment-friendly without producing secondary pollution [134]. Akhbarizadeh et al. synthesized maghemite nanoparticles with the average particle size of 14 nm by using a single-step method and employed them to treat wastewater containing Cu (II), Ni (II), Mn (II), Cd (II), and Cr (VI) by batch method [135]. The results showed that the affinity between maghemite nanoparticles and heavy metals were in the following order: Cu (II) > Cr (VI) > Mn (II) > Ni (II) > Cd (II). Rajput et al. synthesized superparamagnetic maghemite nanoparticles with a tunable morphology by employing a flame spray pyrolysis method [136]. The synthesized maghemite nanoparticles with a surface area of 79.35 m^2^·g^−1^ were employed to remove Pb (II) and Cu (II) in wastewater. The batch adsorption results showed that the maximum Langmuir adsorption capacities were 68.9 and 34.0 mg·g^−1^ for Pb (II) and Cu (II) respectively. Electrostatic interactions were mainly responsible for the metal ions adsorption. The surface of maghemite was covered with FeOH groups in water which could form positive Fe-^+^OH_2_ or negative FeO^−^ groups with the change of pH. More Fe (III) O^−^ or Fe (III) OH sites formed with the increase of pH, thus improving the adsorption capabilities for Pb (II) and Cu (II). While more Fe-^+^OH_2_ sites formed with the decrease of pH, which repelled Pb (II) and Cu (II) on the surface and decreased the removal capacity.

In recent years, reports on polymer-modified maghemite nanomaterials, which combine the superiorities of both polymers and maghemite are gaining increasing attention. Madrakian et al. prepared a novel mercaptoethylamino monomer-modified maghemite nanomaterial (MAMNPs) via synthesis process shown in Scheme 1 [137]. The specific surface area of MAMNPs was 92.41 m^2^·g^−1^ and the maximum removal capacities of Ag (I), Hg (II), Pb (II), and Cd (II) were 260.55, 237.60, 118.51, and 91.55 mg·g^−1^ respectively by using the Sips isotherm. The batch adsorption data could be well represented by the pseudo-second-order kinetic model. Moreover, maghemite nanoparticles have also been reported to be modified by poly (1-vinylimidazole), polyrhodanine, polypyrrole, polyaniline etc. and these polymer-modified maghemite nanoparticles have exhibited good removal capabilities and selectivity towards heavy metal ions [138,139,140].

##### Magnetite (Fe_3_O_4_)

Magnetite (Fe_3_O_4_)-based nanomaterials are another kind of widely used nanometer adsorbent due to their low cost, simplicity of use, easy availability, and environmental friendliness [141]. Similar to maghemite, magnetite-based nanomaterials could be easily separated from the aqueous solution after treatment by adding a magnetic field (Figure 6). There have been numerous reports on their applications in heavy metals treatment [142,143,144]. Giraldo et al. synthesized magnetite nanoparticles by using a co-precipitation method and the obtained nanoparticles were used to treat Pb (II), Cu (II), Zn (II), and Mn (II) in a batch mode [145]. The results demonstrated that nanosized magnetite had the best adsorption effect towards Pb (II) (0.180 mmol·g^−1^) while the least for Mn (II) (0.140 mmol·g^−1^). This difference could result from the diverse electrostatic interactions between the heavy metal ions and the adsorbent sites. The Langmuir isotherm and pseudo-second-order model were found to be correlated with the adsorption data well. It was deduced that the size of hydrated ionic radii may affect the interactions with the negative charged adsorption site. When the hydrated ionic radii increased, the distance to the adsorbing surface would increase and the adsorption would be weaker. Given that Pb (II) had the lowest hydrated ionic radius and the maximum capability to compete with proton, it was reasonable that it had the highest adsorption capacity.

Nevertheless, bare magnetite nanoparticles are easily oxidized with oxygen due to the existence of Fe (II) in their structures, and they also tend to be corroded by acids or bases. Thus, magnetite particles are usually surface-modified by functional groups like –NH_2_ [147], –COOH [148], –SH [149] etc. or coated with a protective shell. Scheme 2 has given a schematic diagram of the core-shell structure based on PI-b-PEG diblock copolymer which encapsulated the single or multiple iron oxide nanoparticles [150]. In this study, FeO_X_-NPs were coated with PI-DETA first and then encapsulated with different amounts of PI-b-PEG diblock copolymer. Baghani et al. synthesized amino functionalized Fe_3_O_4_ nanoparticles by using a simple one-pot method and investigated their adsorption effects towards Cr (VI) and Ni (II) [151]. The adsorption experiments were carried out in a batch mode and the maximum adsorption capacities for Cr (VI) and Ni (II) were 232.51 mg·g^−1^ and 222.12 mg·g^−1^ respectively by calculating from the Langmuir isotherm. The adsorption kinetics followed the pseudo-second-order model and the thermodynamic parameters demonstrated that the adsorption process was endothermic, spontaneous, and entropy favored in nature. The adsorbent after the treatment could be separated from the wastewater in 30 s by adding a magnetic field. As for the coating strategies to form core-shell structure, silica [152], sodium dodecyl sulphate [153], oleate [154], p-nitro aniline [155], polyethylene glycol [156], chitosan [157], tannic acid [158] etc. have been reported to be coated on the magnetite nanoparticles to treat heavy metals in wastewater. For example, Huang et al. synthesized a novel core-shell adsorbent by coating the organodisulfide polymer (PTMT) onto the amino-functionalized magnetite nanoparticles [159]. The newly-synthesized adsorbents showed high adsorption capacities toward high-concentration Pb (II), Hg (II) and Cd (II) in a batch adsorption experiment. When the initial concentrations of these three ions reached 600 mg·L^−1^, the adsorption capacities of Pb (II), Hg (II), and Cd (II) were 533.13, 603.16, and 216.59 mg·g^−1^, respectively. Moreover, the reusability of this adsorbent was also proven to be excellent. The re-adsorption capacities for Hg (II) and Pb (II) remained almost constant even after five cycles of regeneration while the removal capacity for Cd (II) decreased slightly. The modified magnetite nanomaterials with the core-shell structure have shown their great potential for removing heavy metals due to their excellent uptake capacities towards heavy metals and simplicity to be separated from wastewater by taking the advantages of both magnetite core and the organic or inorganic shells. 

##### Hydrous Iron Oxides (HFO)

Hydrous iron oxides (HFO) nanoparticles have shown their great potential for removing heavy metals from wastewater due to their affinity towards heavy metals [160], large surface area [161] and their low cost. The removal mechanism consists of adsorption, ion-exchange and co-precipitation [162]. Though possessing many advantages for treating heavy metals, HFO nanoparticles cannot be used directly in stationary bed or flow-through systems due to their poor mechanical robustness, low hydraulic conductivity, and excessive pressure drop [163]. Thus, HFO are often combined with porous nanomaterials to form composites. For example, a hydrogel-supported HFO nanomaterial HFO-P(TAA/HEA) was prepared and its removal effect towards Pb (II), Cu (II), Cd (II) and Ni (II) from wastewater was investigated in batch mode [164]. The results showed that the removal order of this hybrid nanomaterial in quaternary system were in the following order: Pb (II) > Cu (II) > Ni (II) > Cd (II). The adsorption capacities for these four ions in quaternary system were 0.432, 0.231, 0.1616 and 0.0932 mmol·g^−1^ respectively and they were all higher than that of mere HFO and mere hydrogel. It’s worth noting that the coexistence of Ca (II) was proved to decrease the adsorption capacities of these four heavy metals which may be caused by competitive adsorption. Moreover, HFO nanoparticles were also reported to be an extremely effective adsorbent towards As with the uptake capacity of 92 mg·g^−1^ calculated from Langmuir model [165]. In one study carried out by Huo et al. [166], carboxymethyl cellulose (CMC) modified HFO was synthesized to remove As (V) in the water. The results illustrated that the removal mechanism was mainly comprised of precipitation and surface complexation and the maximum adsorption capacity was 355 mg·g^−1^, which was much higher than the capacities reported in previous literatures. Second-order model and dual-mode isotherm model were successfully used to correlate the sorption kinetics and isotherms of adsorption data in this study. Not limited to the laboratory use, this hybrid HFO-CMC nanomaterial was also employed to treat practical wastewater samples from Realgar mine tailings with an initial As (V) concentration of 38.2 mg·L^−1^. The results indicated that HFO-CMC could remove 90.5% As (V) and the As (V) concentration after the treatment was far below the standard of World Health Organization (WHO). 

#### 3.4.2. Manganese Oxides-Based Nanomaterials

Manganese oxides nanoparticles have also been reported to remove heavy metals from wastewater and nanosized manganese dioxide as well as hydrous manganese oxide (HMO) in which Mn is quadrivalve are mainly discussed here [167].

Nanocrystalline manganese oxide was found to have a high surface area which contributes to its good adsorption performance. Also, M–O^δ+^ and M–O^δ−^ units on the surface of manganese oxide would help the sorption of metal ions [168]. Wang et al. synthesized a novel dumbbell-like manganese dioxide/gelatin and investigated its adsorption performances towards Pb (II) and Cd (II) [169]. The batch adsorption study showed that the maximum adsorption capacities towards Pb (II) and Cd (II) were 318.7 and 105.1 mg·g^−1^ respectively by calculating from the Langmuir model. The adsorption kinetic followed the pseudo-second-order model. What’s more, this nanocomposite could be exploited to treat real water after being supported on an amino-functionalized PMMA plate. High removal capacity, simple operation and good stability implied that this nanocomposite could be applied to treating heavy metals in real water. Besides, nanoscale manganese dioxide could also be used in the adsorption and oxidation of Tl (I) in wastewater [170]. This batch adsorption process could be finished in 15 min and the maximum removal capacity was calculated to be 672 mg·g^−1^ by using Langmuir model, indicating the potential of nanosized manganese oxides to be used in Tl treatment. In addition to these metals, manganese oxides-based nanomaterials have also been reported to treat other heavy metals like U, Cd(II), Cu(II), Pb(II), Zn(II), Hg(II) and so on [171,172,173].

Hydrous manganese oxide (HMO) is another kind of manganese oxides which have shown its advantages in heavy metals removal considering its high surface area, porous structures, and abundant sites for adsorption [160]. The coordination chemistry also plays a significant role in adsorption of HMO because hydroxyl groups on the HMO surface could coordinate with heavy metal ions [174]. The adsorption of heavy metal ions onto HMO was usually comprised of two steps: the quick adsorption of heavy metals onto the external surface and the slow intra-particle diffusion along the micro-pore walls of HMO. Recently, wan et al. prepared a HMO-BC nanocomposite by impregnating the HMO nanoparticles into the biochar (BC) [175]. HMO-BC nanocomposite exhibited an excellent removal effect towards Pb (II) and Cd (II) in a wide range of pH. The combination of HMO and BC could avoid the drawbacks of employing BC individually such as the unsatisfactory adsorption capacity and poor selectivity. Fixed-bed column adsorption experiments demonstrated that the effective treatment capability of HMO-BC for a simulated Pb (II) or Cd (II) containing wastewater was about 4-6 times higher than that of the BC host. Therefore, HMO-BC is a promising alternative for removing heavy metals from the polluted water. 

#### 3.4.3. Zinc Oxides-Based Nanomaterials

Zinc oxide nanoparticles have gained their popularity as adsorbents for heavy metals due to their high surface area, low cost and extraordinary removal capacity [176]. Nanosized zinc oxides have been reported to treat Cr (VI), Cu (II), Ni (II) and so on [176,177,178]. Sheela et al. [179] studied the removal performances of ZnO nanoparticles towards Zn (II), Cd (II) and Hg (II) by using batch method. The results demonstrated that the maximum adsorption capacities were 357, 387, 714 mg·g^−1^ respectively by calculating from the Langmuir model, indicating a highly competitive adsorbent for these heavy metals. The effect of pH was investigated over the range of 4.0–8.0. The result indicated that pH had a significant influence on the adsorption behaviors which could be explained by the surface charge of the ZnO and degree of speciation of sorbents. Besides, a study carried out by Ghiloufi et al. compared the adsorption differences between ZnO nanoparticles and calcium doped ZnO nanoparticles towards Cr (VI), Cd (II)and Ni (II) [180]. The results indicated that the incorporation of Ca in ZnO nanoparticles could improve the uptake effect of these heavy metals. Somu et al. prepared zinc oxide nanoparticles by exploiting casein as the reducing and capping agent [181]. The casein-capped ZnO nanoparticles with an average size of 10 nm were employed to treat wastewater containing three metals and two dyes in batch mode. Adsorption data fitted the Langmuir model well and the adsorption capacities towards Cd (II), Pd (II) and Co (II) were 156.74, 194.93, 67.93 mg·g^−1^ respectively and the uptake capacities towards methylene blue and congo red were 115.47 and 62.19 mg·g^−1^. Besides, this casein-capped ZnO nanoparticles also exhibited excellent antimicrobial activity, indicating they are promising adsorbent towards the practical wastewater.

#### 3.4.4. Titanium Oxides-Based Nanomaterials

Titanium oxides are extensively reported to photodegrade organic pollutants as effective photocatalytic [182]. There are also some reports on their applications on the heavy metal removal [183]. Gulaim et al. had prepared nano-titania with a mesoporous structure via a rapid surfactant-free approach and investigated its adsorption effect towards dichromate with Cr_2_O_7_^2−^ equilibrium concentrations varying from 20–300 mg·L^−1^ [184]. The result indicated that the maximum adsorption capacity of the synthesized TiO_2_ towards Cr_2_O_7_^2−^ was 26.1 mg·g^−1^, which was 12.6 mg·g^−1^ for Cr (VI). This nano-titania showed a higher uptake capability towards Cr_2_O_7_^2−^ than that of the previously reported adsorbent. Youseff et al. synthesized TiO_2_ nanowire with an average diameter of 30–50 nm which was used to remove Pb (II), Cu (II), Fe (III), Cd (II), and Zn (II) from wastewater, and the results showed that 97.06% and 79.77% of Pb (II) and Fe (III) could be removed from the water respectively [185]. Besides, iron-doped TiO_2_ nanoparticles were also synthesized and they exhibited a higher removal capacity towards arsenic than that of pure TiO_2_ nanoparticles. This enhancement was attributed to the termination of grain growth and the response towards visible light [186]. Starch-coated TiO_2_ nanoparticles were prepared to remove Cd (II), Co (II), Cu (II), Ni (II), and Pb (II) in spiked tap-water and the recoveries of these metal ions were all over 90% [187]. Recently, the microwave-enforced sorption (MES) approach was investigated to remove heavy metals from water by using microwave-synthesized TiO_2_-chitosan nanoparticles. MES technique was proven to be a green and rapid technique. The maximum adsorption capacity towards Cd (II) was 1800 μmol·g^−1^. 86.80–88.01% and 72.56–70.67% of Cu (II) and Cd (II) could be removed from water by heating 60–70 s via MES technique [188]. However, the major drawbacks of titanium oxides nanoparticles lie in their complicated production process and separation difficulty [52]. It is usually difficult to separate TiO_2_ nanoparticles after the water treatment when TiO_2_ is implemented in a slurry suspension [189].

#### 3.4.5. Aluminum Oxides-Based Nanomaterials

Aluminum oxides-based nanomaterials are another kind of widely used metal adsorbent towards heavy metals with the advantages of a low manufacturing cost and a high decontamination efficiency [190,191]. Aluminum oxide has several crystalline structures such as α, γ, θ, η etc., among which γ-alumina is the most widely used [192]. γ-Al_2_O_3_ nanoparticles have a great potential as adsorbents attributing to their high specific surface area, excellent adsorption capacity, mechanical strength, and low-temperature modification [193]. Recently, Tabesh et al. synthesized γ-Al_2_O_3_ nanoparticles via a modified sol-gel method and investigated their adsorption effect towards Pb (II) and Cd (II) [194]. The average size of γ-Al_2_O_3_ nanoparticles was 6–13 nm and their maximum removal efficiency towards Pb (II) and Cd (II) achieved 97% and 87% respectively. The adsorption of Pb (II) and Cd (II) on γ-Al_2_O_3_ nanoparticles followed the Freundlich isotherm with an adsorption capacity of 47.08 and 17.22 mg·g^−1^ respectively. Furthermore, a study which investigated the influence of phosphate (PO_4_), citrate, and humic acid (HA) on the adsorption effect of Al_2_O_3_ nanoparticles towards Zn (II) and Cd (II) was also carried out and the result indicated that PO_4_ and HA could promote the adsorption of both Zn (II) and Cd (II), while citrate would decrease the adsorption of Zn (II) in a mono-metal system [195]. In addition to the above heavy metals, Al_2_O_3_ nanomaterials have also been reported to remove Hg (II), As (III), Cu (II), Ni (II), Cr (VI), etc. and have exhibited decent removal capabilities towards these elements [196,197,198,199].

#### 3.4.6. Magnesium Oxides-Based Nanomaterials

Magnesium oxide nanoparticles have many advantages as adsorbents for heavy metals, including an extraordinary adsorption capacity, low cost, nontoxicity, abundance, and environmentally friendly character [200]. Moreover, MgO nanoparticles are also equipped with an excellent antibacterial ability towards Gram-positive and Gram-negative bacteria as well as bacterial spores [201]. Research has indicated that Cd (II), Pb (II), and *Escherichia coli* could be removed from water simultaneously via MgO nanoparticles synthesized by the sol–gel method [200]. Mahdavi et al. investigated the different adsorption effects of nanosized TiO_2_, Al_2_O_3_, and MgO towards Cd (II), Cu (II), Ni (II), and Pb (II) from aqueous solutions [202]. The maximum uptake capacities of TiO_2_ nanoparticles towards Cd (II), Cu (II), Ni (II), and Pb (II) were 120.1, 50.2, 39.3, and 21.7 mg·g^−1^ respectively. The maximum uptake capacities of Al_2_O_3_ nanoparticles towards these four ions were 118.9, 47.9, 35.9, and 41.2 mg·g^−1^ while the adsorption capacities of MgO nanoparticles were 135, 149.1, 149.9, and 148.6 mg·g^−1^. The removal mechanism of MgO nanoparticles towards these four metals was attributed to adsorption and precipitation while the removal mechanisms of TiO_2_ and Al_2_O_3_ were mere adsorption. Tatenda C. et al. synthesized MgO nanoparticles via a combustion method and investigated their removal capability towards Cu (II) [203]. The results indicated that 96% of Cu (II) could be removed from a 10 ppm CuCl_2_ solution by using 0.2 g MgO nanoparticles, while the commercial MgO only exhibited a removal efficiency of 15%. Xiong et al. reported that the maximum adsorption capacities of MgO nanoparticles towards Cd (II) and Pb (II) were 2294 and 2614 mg·g^−1^ respectively, by calculating from the Langmuir equation in a batch adsorption experiment [204]. The adsorption process was influenced simultaneously by external mass transfer and intraparticle diffusion. The high adsorption capacities were attributed to the generation of OH^−^ dissociated from Mg (OH)_2_, which was hydrated from MgO and the synergistic effects of precipitation and adsorption helped to achieve this adsorption capacities. In another batch adsorption study carried out by Feng et al., mesoporous MgO nanosheets were prepared (Scheme 3.) and they exhibited extraordinary removal capacity towards Ni (II) with the value of 1684.25 mg·g^−1^ calculated from the Langmuir model [205]. The pseudo-second-order model fitted the adsorption kinetic well. Distillation treating process could raise the S_BET_ of MgO nanosheet up to 181.692 m^2^·g^−1^. Therefore, MgO nanoparticles have exhibited a promising application potential for removing heavy metals from wastewater.

#### 3.4.7. Cerium Oxides-Based Nanomaterials

Nanosized cerium oxide (CeO_2_), a non-harmful rare-earth oxide in which Ce is quadrivalve, have been applied in many areas such as photocatalysis and sensing [206], UV blocking [207], water treatment, etc. [208]. The crystalline size, bulk density, porosity, and surface area of CeO_2_ were reported to have great influence on their activity, stability, dispersion behavior, and selectivity, thus influencing the effect of heavy metals removal. Recillas et al. investigated the adsorption effect of CeO_2_ nanoparticles towards Cr (VI) from water [209]. The synthesized CeO_2_ nanoparticles had an average size of 12 nm and surface BET area of 65 m^2^·g^−1^. The maximum adsorption capacity for Cr (VI) was 121.95 mg·g^−1^ in this work with the initial Cr (VI) concentration of 80 mg/L, indicating a good choice for removing low amounts of Cr (VI) from water. CeO_2_ nanoparticles with an average size of 3–5 nm and surface area of 257 m^2^·g^−1^ were synthesized by Mishra et al. and they were used to remove As (III) and As (V) in water by the batch method [210]. The adsorption capacities towards these two ions were 71.9 and 36.8 mg·g^−1^ respectively, as determined from the Langmuir isotherm and this adsorption process was almost complete after 10 min. It is noteworthy that the coexistence of anions such as H_2_PO_4_^−^, SO_4_^2−^, and HCO_3_^−^ would bring down the adsorption capacity. Moreover, CeO_2_ has also been reported to cooperate with other metal oxides to treat heavy metals [211,212]. Different morphologies of CeO_2_ nanopowder were prepared and the removal effect of samaria-doped ceria nanopowder (SDC) towards Pb (II), Cu (II), and Zn (II) was investigated [213]. The results showed that spherical SDC nanopowder (SDC-F) had a higher adsorption capacity than that of cluster plate-like SDC nanopowder (SDC-I). 

#### 3.4.8. Zirconium Oxides-Based Nanomaterials

Nanosized zirconium oxides are another kind of promising metallic oxides adsorbent which can be used to remove heavy metals in wastewater. Their advantages are that they have plenty of -OH on their surfaces and possess large surface areas. Moreover, nanosized zirconium oxides own great chemical stabilities and exhibit excellent adsorption affinities towards heavy metals like Pb (II), Zn (II) and Cd (II) [214]. Zirconium oxides nanomaterials discussed in this chapter are mainly comprised of nanosized zirconia and hydrous zirconia (HZO) based nanomaterials.

Gulaim et al. investigated the removal effects of a series of transition-metals which had mesoporous structures towards Cr (VI) in solutions, including TiO_2_, ZrO_2_, HfO_2_, Nb_2_O_5_, and Ta_2_O_5_ [215]. The synthesized metal oxides all exhibited large surface areas and were comprised of partially combined homogeneous nanoparticles with crystalline cores and amorphous shells. The adsorption capacity of ZrO_2_ for Cr (VI) was 73.0 mg·g^−1^, indicating that mesoporous structure ZrO_2_ was an attractive adsorbent for Cr (VI). Yalçınkaya et al. synthesized a novel nanocomposite ZrO_2_/B_2_O_3_ and investigated its removal effects towards Co (II), Cu (II), and Cd (II) by column adsorption [216]. The adsorption capacities for Co (II), Cu (II), and Cd (II) were 32.2, 46.5, and 109.9 mg·g^−1^ respectively by using Langmuir model and the reusability of this nanocomposite proved to be satisfactory. Hybrid nanocomposite ZrO_2_/B_2_O_3_ has provided a simple, economical, and selective method for the separation of Co (II), Cu (II), and Cd (II).

Nanosized hydrous zirconia (HZO) has also shown its great potential as adsorbent for heavy metals. Zhang et al. synthesized polystyrene-supported nanosized zirconium hydroxide HZO-PS and studied its removal effect towards Cd (II) [217]. The result indicated that Cd (II) could be removed by using this nanocomposite in a wide pH range varying from 2.5–7.0 with negligible Zr releases. Fixed-bed column adsorption experiment was also conducted and the results indicated that this nanocomposite has good applicability with the treated capacity of 750 bed volume per run. In another study carried out by Hua et al., HZO was combined with a commercial cation exchange resin D-001 to make a nanocomposite NZP [218]. The removal effects of NZP towards Pb (II) and Cd (II) were investigated by column adsorption method and the result demonstrated that the experimental maximum adsorption capacities for these two ions were 319.4 and 214.7 mg·g^−1^ respectively. Pseudo-first-order model was found to fit the adsorption kinetics better. The cyclic column experiments showed that the synthesized NZP could be used to treat both synthetic and real acid mine wastewater repeatedly without any capacity loss when the HNO_3_−Ca (NO_3_)_2_ solution served as the regenerant.

### 3.5. Nanocomposite Nanomaterials

Despite the fact that each kind of nanomaterial discussed above has their own advantages, their respective drawbacks cannot be neglected. For example, it is difficult for CNTs to suspend uniformly in different solvents, while nZVI are prone to get oxidized [219]. Furthermore, nanoparticles often have the problem of aggregation, poor separation, and an excessive pressure drop when used in fixed-bed and flow-through systems [26,220]. A common strategy to solve these problems is to synthesize hybrid nanocomposites which take the advantages of different nanomaterials [109]. In this part, nanocomposites of inorganic and organic polymer supports, together with the magnetic nanocomposites are intensively discussed.

#### 3.5.1. Inorganic-Supported Nanocomposites

Inorganic supports of nanocomposites which are used for heavy metals removal are mainly consisted of activated carbon (AC), CNTs, and some natural materials such as bentonite, montmorillonite, zeolite, and so on. AC is one of the most effective, economic and simplest adsorbent for pollutants in the aqueous solutions [221]. There have been some reports on the AC-supported nanocomposites which were used to remove heavy metals from water and these composites exhibited great potential for removing Cr (VI), Pb (II), Cd (II) etc. [222,223,224,225]. CNTs-supported nanocomposite is one kind of nanomaterial mainly supported on CNTs. Chitosan is one of the most widely used polymer to modify CNTs to prepare this kind of nanocomposites. Salam et al. prepared multi-walled CNTs/chitosan nanocomposite by sonicating the chitosan and CNTs suspension and then crosslinking them with glutaraldehyde (Figure 7) [226]. The composite was packed into a glass column to remove Cu (II), Zn (II), Cd (II), and Ni (II) from aqueous solution (Figure 8) and it exhibited high removal efficiency towards the targeted metal ions. Besides, polyethylenimine, 3-mercaptopropyltriethoxysilane, 8-hydroxyquinoline, cyclodextrin etc. have been reported to hybridize with CNTs to make nanocomposites [227,228,229,230]. Recently, Hayati et al. synthesized CNT coated poly-amidoamine dendrimer (PAMAM) nanocomposite (CNT/PAMAM) and investigated its adsorption efficiency towards As (III), Co (II), and Zn (II) in a fixed-bed system [231]. The maximum uptake capacities towards these three ions were 432, 494, and 470 mg·g^−1^ respectively in column adsorption studies. Bentonite is one of the most potential candidates for treating high-concentration heavy metal pollution due to its excellent properties such as high specific area, cation exchange ability and adsorptive affinity [232]. NZVI, magnetite, hexadecyltrimethylammonium bromide (CTMAB), ethylene diamine tetraacetic acid (EDTA), 2-mercaptobenzothiazole (MBT), cellulose etc. were reported to be combined with bentonite and employed to remove heavy metals from aqueous solution [30,103,233,234]. Zeolite is another kind of promising host and stabilizer for nanoparticles due to its high surface area, excellent ion exchange ability, hydrophilic, environment-friendly characteristic and simply controllable chemical properties [235]. Hydroxyapatite/zeolite nanocomposite (HAp/NaP) was prepared to treat Pb (II) and Cd (II) in water by batch method and the maximum adsorption capacities of Pb (II) and Cd (II) were 55.55 and 40.16 mg·g^−1^ respectively [236]. The adsorption kinetic could be correlated with the pseudo-second model perfectly. HAp/zeolite nanocomposite also exhibited antibacterial activity towards most common Gram-positive and Gram-negative bacteria, showing its application potential in water treatment. There are still many other inorganic supports of nanocomposite such as GO, sand, clay, etc., and they were found to be promising alternatives to remove heavy metals from wastewater [24,237,238].

#### 3.5.2. Organic Polymer-Supported Nanocomposites

Polymeric hosts have many extraordinary properties, such as excellent mechanical strength, tunable functional groups, feasible regeneration, environmental soundness, and a degradable characteristic which make organic polymers a competitive option of hosts for nanocomposites [239]. Polymer-supported nanocomposites are comprised of two types, synthetic organic polymer-supported nanocomposites and biopolymer-supported nanocomposites [240]. The fabrication of polymer nanocomposites could be achieved by two ways, direct compounding and in situ synthesis, which are illustrated in Scheme 4 [109]. The synthetic organic polymer, such as polystyrene (PS), polyaniline, polyaniline (PAN), etc. [76,241,242], have been widely reported to fabricate nanocomposites which are exploited to treat heavy metals. For example, Afshar et al. fabricated polypyrrole-polyaniline/Fe_3_O_4_ magnetic nanocomposite and investigated its removal capacity towards Pb (II) in an aqueous solution [243]. The synthesized nanocomposite could remove almost 100% of Pb (II) at pH = 8–10 when the concentration of Pb (II) was 20 mg·L^−1^. Freundlich isotherm model and the pseudo-second order model fitted the adsorption data better.

Apart from the synthetic organic polymers, biopolymers like cellulose, chitosan, alginate, etc. are also extensively used as supports of nanocomposites. Cellulose, as one of the most common biopolymers, possesses hydroxyl groups on its glucose ring which provide abundant coordination sites for heavy metal ions. Thus, it is a promising starting material of adsorbents [244]. Suman et al. developed a nanocellulose (NC)-Ag nanoparticles (AgNPs) embedded pebbles-based composite material which was employed to remove dyes, heavy metals and microbes in water by column adsorption method [245]. The results indicated 99.48% of Pb (II) and 98.30% of Cr (III) were removed from water along with a 99% decontamination efficiency for microbial load. Chitosan is another eco-friendly and biodegradable material which has great potential for the adsorption of heavy metals due to the coexistence of –NH_2_ and –OH in its structure. Saad et al. synthesized ZnO/chitosan core-shell nanocomposite (ZOCS) which possessed the advantages of low cost and less biological toxicity and investigated its removal capabilities for Pb (II), Cd (II) and Cu (II) [246]. The batch adsorption results demonstrated that the maximum adsorption capacities for Pb (II), Cd (II), and Cu (II) were 476.1, 135.1, and 117.6 mg·g^−1^ respectively according to the Langmuir isotherm model and this nanocomposite could be repeatedly used with an excellent adsorption capacity. Alginate, which is extracted from brown seaweed, is a non-toxic, biocompatible, and biodegradable biopolymer [247]. Gokila et al. fabricated the chitosan/alginate nanocomposite in order to remove Cr (VI) from wastewater [248]. The maximum adsorption capacity of Cr (VI) in the batch adsorption experiment was 108.8 mg·g^−1^, and the adsorbent favored a multilayer adsorption. Lofrano et al. gave a detailed review on polymer functionalized nanocomposites (PFNCs) for the removal of metals from water, covering the preparation, characterization, toxicity, removal capabilities of PFNCs, and the interactions between the polymer hosts and nanoparticles [249]. Though the polymer-based nanocomposites have shown their great potential for heavy metals removal, their synthesis process, costs, recovery techniques, safety, etc. still need more investigation.

#### 3.5.3. Magnetic Nanocomposites

Magnetic nanocomposites are one type of peculiar nanomaterial which have been receiving increasing attention due to their easy separation ability. Magnetic nanocomposites are mostly based on magnetic iron and iron oxides. The fabrication of these magnetic nanocomposites could be achieved mainly through three approaches: (1) Surface modification of magnetic iron/iron oxide nanoparticles by functional groups such as –NH_2_, –SH etc., which has been discussed in the previous chapter; (2) encapsulating the iron/iron oxide nanoparticles with other materials, such as humic acid, polyethylenimine, polyrhodanine, MnO_2_, polypyrrole, etc. to make a core-shell structure [139,172,250,251,252]; (3) coating the iron/iron oxide nanoparticles on some porous materials such as graphene oxide, CNTs, and so on [253,254]. Recently, Huang et al. synthesized a novel magnetic composite in which the nanosized Fe_3_O_4_@SiO_2_ served as the core while the amino-decorated Zr-based metal-organic frameworks (Zr-MOFs) acted as the shell [255]. The amino-decorated metal-organic frameworks composites exhibited an efficient removal capacity towards both Pb (II) and methylene blue. Ge et al. fabricated a Fe@MgO nanocomposite (Scheme 5) which combined the advantages of nZVI’s strong magnetism and MgO’s high adsorption capacity [256]. The maximum removal capacities of this nanocomposite towards Pb (II) and methyl orange were 1476.4 and 6947.9 mg·g^−1^ respectively in batch adsorption experiments, indicating a great superiority for water treatment. The adsorption data of Pb (II) and methylene orange fitted the Langmuir model and the pseudo-second-order kinetic well, indicating that the adsorption onto the absorbent was a monolayer chemisorption. 

It is evident that magnetic nanocomposites possess great potential for the removal of heavy metals from water due to their easy separation property. However, the biocompatibilities of magnetic nanocomposites should be considered for a further adhibition.

## 4. Conclusions and Perspectives

Nanomaterials have been extensively exploited to remove heavy metals in water owing to their exceptional properties. In this work, a series of nanomaterials, including carbon-based nanomaterials, zero-valent metal nanomaterials, metal oxide materials, and nanocomposites were discussed in detail. These materials are summarized in Table 3 at the end of this review. These nanomaterials exhibit great advantages as adsorbents towards heavy metals.

Nevertheless, there are still some bottlenecks that needed to be overcome to make better use of these nanomaterials in water treatment. First, most nanomaterials are unstable and tend to aggregate, thus reducing their removal capacity. Furthermore, it is usually difficult to separate the nanomaterials from the aqueous solution swiftly and efficiently due to their nanoscale size. The proposal of nanocomposites seems to be a promising approach to solving these problems. However, the synthesis process, long-term performance, and some other issues correlated with nanocomposites need further investigation. Second, the commercial nanomaterials used for heavy metals removal on an industry scale are rare and more efforts are needed to develop market-available nanomaterials. The synthesis, as well as operating costs of nanomaterials should be optimized for the sake of the economy and the production of these nanomaterials should meet the requirements of green chemistry. Last but not least, with the increasing use of nanomaterials in waste water treatment, their impacts and toxicities towards both the environment and human beings should be taken into consideration. Although there have been some literatures focused on the toxicity and biological behavior of nanoparticles towards human health [257,258], the biocompatibility of nanomaterials with the environment also needs to be further investigated due to the insufficient criteria on the toxicity of nanomaterials nowadays. Regulations on the employment of nanomaterials are highly demanded in order to reduce the adverse effects of nanomaterials towards both environment and human beings. 

In all, the treatment of heavy metal in wastewater is of great significance from the perspective of both the ecological environment and human health. The emergence of nanomaterials has provided us with a promising alternative to the traditional adsorbents for removing heavy metals. However, the removal capabilities of these nanomaterials are mostly investigated in stimulated water with relatively simple components. The reports of their use in practical wastewater are insufficient and are highly in need. Additionally, the interaction mechanisms between the hosts and guests of the nanocomposites demand a thorough understanding in order to guide the synthesis of nanocomposites. The risk and impacts of nanomaterials cannot be neglected when we develop them. There is still a long way to go to put nanomaterials into practical heavy metal water treatment, in particular when considering comprehensively their removal capability, reusability, separation, synthesis, and cost.

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
