# Peer review of "Nanomaterials for the Removal of Heavy Metals from Wastewater"

_nanomaterials, 2019, doi:10.3390/nano9030424_

Reviewer 1 Report

The proposed review presents a summary of techniques applying nanomaterials for wastewater purification. The topic is highly important and some aspects are reflected well in the proposed manuscript, especially the use of zerovalent metal particles. The overview of the use of metal oxides is rather incomplete, missing recent progress in mesoporous matrices of TiO2, ZrO2 etc. (see J. Mater. Chem. 2012, 22, 20374-20380 and Chem Eur. J. 2014, 20, 10732-10736). Surface-functionalized metal oxides have also been used for removal of potentially radioactive heavy metal pollutants, see RSC Adv. 2015, 5, 24575-24585.

What is especially important to complement to this review with is a chapter on nano silica based materials. The few references to FeOx@SiO2 provided in this review have to be placed into this chapter as corresponding to magnetic silica materials. Silica-based functional nanoadsorbents for wastewater and drinking water treatment have been actively developed by groups of Strelko and especially Zub et al. Some clues can be found for stabilization of magnetic nano silica in RSC Adv. 2014, 4, 22606-22612. Silica adsorbents bearing surface functions have been described in Colloids & Surf. A 2014, 259, 4-10; Applied Surf Sci. 2017, 420, 782-791; Beilstein J. Nanotech. 2017, 8, 334-347; Sci Rep. 2018, 8, 8592, 1-13. Combination of heavy metal removal with removal of  organic pollutants by combination of molecular and biocatalytic functions on nano silica have been reported in Coll. Surf. B 2016, 144, 135-142; Nanomaterials 2017, 7, 298, 1-17; ACS Sustan. Chem Eng. 2018, 6, 9979-9989.

Additionally, there is a problem with a reaction equation on page 5. It shows surface function on iron oxide surface as peroxide, =Fe-OOH instead of hydroxide, =Fe-OH and is not charge balanced.

Author Response

Reviewer #1:

Point 1: The overview of the use of metal oxides is rather incomplete, missing recent progress in mesoporous matrices of TiO2, ZrO2 etc. Surface-functionalized metal oxides have also been used for removal of potentially radioactive heavy metal pollutants 

Response 1: Actually, nanosized TiO2 materials have been discussed in chapter 4.4 in my first version, though the mesoporous TiO2 was not discussed. Now Gulaim et al’s work has been supplemented in this chapter considering your suggestions. It is really a pity that I omit the applications of nanosized ZrO2 as adsorbents for heavy metals, they have also been added in chapter 5.8. As for the removal of radioactive heavy metal pollutants, it is regrettable that they were not intensively discussed in my review. But the removal for uranium has been discussed in chapter 5.1.1. Thanks for your kind suggestions.

Point 2: What is especially important to complement to this review with is a chapter on nano silica based materials. The few references to FeOx@SiO2 provided in this review have to be placed into this chapter as corresponding to magnetic silica materials. Silica-based functional nanoadsorbents for wastewater and drinking water treatment have been actively developed by groups of Strelko and especially Zub et al. Some clues can be found for stabilization of magnetic nano silica in RSC Adv. 2014, 4, 22606-22612. Silica adsorbents bearing surface functions have been described in Colloids & Surf. A 2014, 259, 4-10; Applied Surf Sci. 2017, 420, 782-791; Beilstein J. Nanotech. 2017, 8, 334-347; Sci Rep. 2018, 8, 8592, 1-13. Combination of heavy metal removal with removal of  organic pollutants by combination of molecular and biocatalytic functions on nano silica have been reported in Coll. Surf. B 2016, 144, 135-142; Nanomaterials 2017, 7, 298, 1-17; ACS Sustan. Chem Eng. 2018, 6, 9979-9989.

Response 2: Thanks for your suggestion for complementing the silica-based nanomaterials in my article which I hadn’t taken into consideration before. I have already supplemented these contents in chapter 3. Some references you recommended have been cited. Though the few references to FeOx@SiO2 were not placed in this chapter, they are put in the chapter of magnetic nanomaterials. The silica-based nanomaterials which could remove heavy metals and organic pollutants have also been reported in this chapter. Thanks for your helpful suggestions.

Point 3: There is a problem with a reaction equation on page 5. It shows surface function on iron oxide surface as peroxide, =Fe-OOH instead of hydroxide, =Fe-OH and is not charge balanced.

Response 3: The oxide shell of nanosized zero-valent iron could be expressed as FeOOH, which was discussed in Li’s work (J. Phys. Chem. C 2007, 111, 6939-6946).  In aqueous phase, Fe2+ is first formed at the surface and rapidly oxidized to Fe3+, which further reacts with OH- or H2O to form hydroxide or oxyhydroxide:                                      

Subsequent dehydration generates FeOOH:

As for the reaction equation (2), the stoichiometric reaction equation should be expressed as follows:

Reviewer 2 Report

The review is well organized.

Issues with the use of nanoparticles for decontamination is rather conflicting as the secondary contamination cannot be ruled out. There are some kinds of nanoparticles, which are not included in this review. 

Some groups of materials perform well or even better when in the composite form.

However, the materials mentioned in this review are well discussed.

Author Response

Reviewer #2:

Point 1: Issues with the use of nanoparticles for decontamination is rather conflicting as the secondary contamination cannot be ruled out. There are some kinds of nanoparticles, which are not included in this review.  

Response 1:  Indeed, issues with the use of nanoparticles for decontamination is rather conflicting as the secondary contamination cannot be ruled out. But they have provided us a new approach to treating heavy metal wastewater and their advantages cannot be neglected. The aim of this article is to summarize the widely reported nanomaterials for heavy metals removal. Due to the limitations of my own scope and knowledge, it’s a pity that I omitted some nanoparticles. Some supplementations to this review have been made as you suggested. Silica-based nanomaterials and zirconium oxides-based nanomaterials have been added in this review. Thanks for your helpful suggestion. But I am sorry to bother you to reread this review to check whether my complementation is appropriate.

Reviewer 3 Report

Dear Editor,

The manuscript (nanomaterials-443960) reviews various nanomaterials studied for the sake of adsorption thus removal of heavy metal ions from water. The idea is interesting and the paper has a good outline and is well-written. However, there are several major comments that need to be addressed before publication, including:

1- No relevant review papers have been cited and compared in terms of perspective and scope. For instance: Farag et al., J Environ Anal Toxicol 2012, 2:7(http://dx.doi.org/10.4172/2161-0525.1000154)

2- The emphasis in this paper is mainly on nanoparticles rather than 1D (nanofibers) and 2D (nanosheets) nanomaterials. I suggest the authors to include some discussions about such kind of nanomaterials as well. With respect to graphene and metal oxide nanomaterials for water treatment, several related review papers are available that could be cited e.g. Nanomaterials 2017, 7(11), 383, Nano Lett., 2016, 16 (4), pp 2860–2866, NPG Asia Materials 9 (8), e427.

3- Introduction can be enriched with some more information regarding the importance of water treatment, water scarcity problem, heavy metals release sources and the potential health problems with such contaminants.

4- Page 4, line 135-136, I suggest the authors to include a table summarizing various developed systems based on Graphene that have been studied for the sake of heavy metal removal. This idea can be applied to other adsorbents discussed in this paper, as well.

5- Page 9, line 303, the adsorption mechanisms in the various systems introduced must be discussed in more details and more scientifically, a simple phrase of “electrostatic interaction” with no mentioning the type of the counteracting groups on the adsorbent is insufficient. This problem is repeated many times, including page 10, lines 326-328, Page 12, lines 361-363.

6- The adsorbent systems should be discussed in terms of kinetics, isotherms and thermodynamics as well. Relevant models such as Lagergen, Freundlich can be referred for explanation of such behaviors.

7- Page 12, scheme 2 needs to be enriched with more information and the graphical elements must be defined.

8- Page 12, line 382, WHO should be defined fully.

9- Some Figures are not informative and irrelevant. e.g. Figure 7,8

10- Frequently surface area is introduced as the main cause for adsorption efficiency of a nanosystem which is extremely non-convincing. Chemistry plays a significant role that needs to be considered. For instance, Page 13, lines 392, and 405.

11- The adsorption test mode must be mentioned, batch or dynamic?

Author Response

Reviewer #3:

Point 1: No relevant review papers have been cited and compared in terms of perspective and scope. 

Response 1: Actually, some relevant review papers were cited in my first version (Ref [2,17,18]) in the introduction part, though I did not compare them in detail. Thanks to your suggestion, some complements have been made in this part. The review paper you suggested has been cited in the introduction.

Point 2: The emphasis in this paper is mainly on nanoparticles rather than 1D (nanofibers) and 2D (nanosheets) nanomaterials. I suggest the authors to include some discussions about such kind of nanomaterials as well.

Response 2: 1D and 2D nanomaterials have been mentioned in this review, including CNTs, graphene-based nanomaterials, TiO2 nanowires, MgO nanosheets. Indeed, the 1D and 2D nanomaterials were still insufficient in this review, so I have made some supplements in this review. The report on graphene-based microbots have been included in this paper.

Point 3: Introduction can be enriched with some more information regarding the importance of water treatment, water scarcity problem, heavy metals release sources and the potential health problems with such contaminants.

Response 3: Thanks for your helpful suggestion. The introduction part has been enriched as you suggested. Some contents related to the importance of water treatment, water scarcity problem, heavy metals release sources and the potential health problems with such contaminants have been added.

Point 4: Page 4, line 135-136, I suggest the authors to include a table summarizing various developed systems based on Graphene that have been studied for the sake of heavy metal removal. This idea can be applied to other adsorbents discussed in this paper, as well.

Response 4: The table named Table 1. which summarized the graphene-based nanomaterials that could be used to remove heavy metals from wastewater has been added in this chapter as you suggested. It’s a pity that this idea isn’t applied in other parts, though a table which involved the nanomaterials in this review has been given at the end of the review.

Point 5: Page 9, line 303, the adsorption mechanisms in the various systems introduced must be discussed in more details and more scientifically, a simple phrase of “electrostatic interaction” with no mentioning the type of the counteracting groups on the adsorbent is insufficient. This problem is repeated many times, including page 10, lines 326-328, Page 12, lines 361-363

Response 5: We felt sorry that we did not put the mechanisms in clear and some revisions have been made. For the adsorption mechanisms in page 9, line 303 and page 10, lines 326-328 as you mentioned, detailed explanations about the mechanisms have been made. But as for the mechanism in page 12, line 361-363, complements were not made to our regrets because it didn’t appear in a specific case. Our aim was only to inform that the adsorption mechanisms of HFO could be consisted of those items.

Point 6: The adsorbent systems should be discussed in terms of kinetics, isotherms and thermodynamics as well. Relevant models such as Lagergen, Freundlich can be referred for explanation of such behaviors.

Response 6: Thanks very much for your suggestions. Adsorption kinetics, isotherms and thermodynamic were omitted due to carelessness. Complements about the adsorption kinetics, isotherms and thermodynamics in most cases have been made as you suggested.

Point 7: Page 12, scheme 2 needs to be enriched with more information and the graphical elements must be defined.

Response 7: An explanation about scheme 2 has been given, although I am not sure what enrichment should be done on scheme 2, maybe you can kindly give me some hints.

Point 8: Page 12, line 382, WHO should be defined fully.

Response 8: WHO has been defined in my revised manuscript.

Point 9: Some Figures are not informative and irrelevant. e.g. Figure 7,8

Response 9: Figure 7,8 has been deleted from my first manuscript considering your helpful suggestions.

Point 10: Frequently surface area is introduced as the main cause for adsorption efficiency of a nanosystem which is extremely non-convincing. Chemistry plays a significant role that needs to be considered. For instance, Page 13, lines 392, and 405.

Response 10: Thanks for your helpful suggestions. Indeed, surface area is not the determinant of the removal capabilities of these nanomaterials. Revisions have been made considering your suggestion. The role of chemistry plays in the adsorption has been introduced in these two places you referred to.

Point 11: The adsorption test mode must be mentioned, batch or dynamic?

Response 11: Given that most of the adsorption test mode was batch in this review, the adsorption mode was not mentioned in the original manuscript. But now the supplements on the adsorption mode of the adsorption tests have already been made. Thanks very much for your helpful suggestions.

Reviewer 4 Report

The review is pleasant to read but I think it requires substantial editing.

Please find here a list of remarks and things to fix:

Line   --->   Comment

10-11   --->   First statement of the abstract is of little usefulness. Better start directly with “Removal of contaminants…”

11-16   --->   This part should be improved reducing the redundant terms (“heavy metals” and “wastewater” are mentioned 3 times each in 6 lines)

29   --->   Why biodegraded? Metals cannot be “degraded” like organic compounds. Please specify

32-36   --->   In this review, MOFs have been excluded, however they should be mentioned at least here. For example these two citations should be added:

DOI: 10.1016/j.jhazmat.2012.11.011

DOI: 10.1039/C5TA04154F

Many more can be found in literature

45-47   --->   This sentence needs a reference

52   --->   Change suggestion: “various kind of nanomaterials, including” > “the following nanomaterials:”

55   --->   I think this sentence should have “are” in place of “were” (present)

57-58   --->   This sentence needs a reference

59   --->   What do you mean with “simplicity”? Please detail.

68   --->   Typo: “muti” -> “multi”. In the same line CNTs are mentioned as “graphene materials”, but this is not correct. I suggest to change with “carbon-based materials”

71   --->   Please provide a range of m2/g and mg/g regarding surface area and adsorption capacity

77   --->   In the caption of Figure 1, a bundle is mentioned, but this is not referred in the main text. Please fix the incongruence

80   --->   Which functional groups? Please mention some meaningful ones.

84-88   --->   I do not get here where is the novelty. Please reformulate if necessary

99   --->   Is this cost issue still valid in 2019? Please check the market and correct if needed.

114   --->   Which functional groups? Please mention some meaningful ones.

116   --->   Please provide a range of m2/g regarding surface area

117   --->   Which “ample” (?) functional groups? Please mention some meaningful ones.

126   --->   Where does the humic acid come from?

128   --->   This phrase says nothing, there should be a follow up, for example “…higher than the commonly reported adsorbents, such as…”

138   --->   I think the term GOCA is not correct referred only to calcium alginate. Please check.

146   --->   Change suggestion: “are still under the lab level” > “are still at a preliminary stage of research” or “are not commercialized yet”

152   --->   Typo: “meal” > “metal”

176   --->   Addition suggested: “The standard potential E0 of some environmentally relevant metals…”

234   --->   Is high temperature OR pressure, or high temperature AND pressure? Please check and add a reference.

241-243   --->   Please cut this in two statements

243   --->   Nanosized is repeated twice in the same line. Please rephrase

244   --->   Although for some elements the state of oxidation is trivial (Al (III), Mg(II)), for others is not. Please explicit the oxidations state of iron, manganese and cerium relevant for this review.

247-251   --->   This part is not much meaningful and it should be expanded a little more and enriched by some references, especially for the first two sentences.

253   --->   Change suggestion: “often exist in the form of minerals in the nature” > “already exist as minerals in nature”. Moreover, it would be better to out “is” instead of “was” (present)

254   --->   It would be more preferable the form “thanks to” rather than “as a result of” in this part.

256   --->   I suggest to describe uranium in full rather than U to avoid confusion. Please add the uptake of uranium from the reference.

266   --->   Although Kelvin is the SI standard, please add or convert to Celsius for the readers.

270   --->   Please fix this phrase (“Hematite is highly resistant to corrosion and is the most stable…”)

292   --->   Please provide a range of m2/g regarding surface area

293   --->   This sentence needs a reference

300   --->   The term “regulable” does not exist. Please change with “tunable”

320   --->   Change: “employ” > “use”

321   --->   Change suggestion: “More importantly” > “Similarly to maghemite”, as maghemite and magnetite are structurally analogous and both have magnetic properties. In this regard, the authors should specify which and where they are talking about superparamagnetic and/or ferromagnetic systems.

326-328   --->   Is this phrase coming from a reference or from the authors? Please motivate the claim (with bibliography).

336   --->   I suggest to put the terms “acid” and “base” as plurals

338   --->   Change: “protection” > “protective”

351   --->   Please report the number of cycles, to support the reusability of the adsorbent.

367-376   --->   In this section, the authors should include the adsorption capacities of the mentioned systems (mg/g).

409   --->   How small are these micropores? Please add the related values.

419   --->   It looks a bit dodgy that a Zn based system is used to uptake Zn. Please check and correct if necessary

421   --->   Please fix and report the pH range

436   --->   Please report the nanowire diameter.

440   --->   The term “arrestment” sounds weird. Perhaps “stop” or “termination” are more appropriate.

443-445   --->   Please rephrase this statement as it is unclear.

447-449   --->   Can you comment more on this sentence?

457   --->   This is too generic, the authors can focus directly on the gamma phase.

458   --->   Change to: “which is the most widely used”

470-471   --->   Change suggestion to: “…have also been reported to remove Hg(II),…”

477   --->   Bacteria names are usually written in italics.

481, 488, 492, 496, 537, and 649   --->   Please check and correct the spacing between words.

505   --->   Change to: “photocatalysis”. The following “water treatment” requires a reference.

515   --->   Change suggestion: “could be finished in several minutes” > “is complete after (NUMBER) minutes”

521   --->   Please explain the terms SDC-F and SDC-I

525-528   --->   This sentence needs a reference

529   --->   This sentence needs a reference. Moreover I suggest to change “were” with “are” (present)

537-538   --->   Please rephrase this statement as it is unclear.

543   --->   Please check the terms “thiol” and “dendrimer” as they look to be placed without a rationale.

546-547   --->   Unnecessary duplications should be avoid, please rephrase

550-552   --->   Here there is a list of many things but some systems looks to be the core, some systems the coating. It is unclear, please check and rephrase.

578   --->   Which nanocomposite(s) are the authors referring to? What is the core/inorganic part involved here?

606   --->   The term “poisonousness” sounds weird. Perhaps “toxicity” is more appropriate.

613, 614, 616, and 618   --->   What does ferric/ferric oxide mean? Maybe the authors refers to iron and iron oxide? Please fix these terms.

632   --->   The high cost of magnetic nanoparticles is fairly debatable nowadays. If so, please state some figures in regard and compare with current adsorbents.

638   --->   Change “have exhibited” with “exhibit”

641   --->   Please define this difficulty, as there are many standard processes that can be used, such as filtration, centrifugation, flocculation, which are possible also at the industrial scale.

646   --->   Change “need to be paid” with “are needed”

653-655   --->   What do the authors mean with “it is high time”? Please rephrase.

656-661   --->   The last sentence is not enough and too general, the authors should put more efforts in providing a more meaningful closure.

662   --->   A summary table listing all the uptakes mentioned in this review, with the related literature references, is helpful for the reader and is strongly recommended.

Author Response

Tianjin University

School of Chemical Engineering and Technology, Tianjin University, Tianjin 300072, P. R. China

February, 9th, 2019

Dear Prof. Tracy Jin:      

Thank you very much for your patience and for giving us the opportunity to further revise our above manuscript. The reviewers' comments as well as your suggestions on our manuscript have been received recently and they are greatly helpful for us to improve the quality of our manuscript. We are pleased to have the opportunity to revise our manuscript and submit a revision. Some changes based on the comments and suggestions are outlined on the attached pages point by point, and the revised manuscript is enclosed in the attachment in which the revised portions are marked in red.

We hope these revisions are to your and the reviewer's satisfaction. If any further clarification or modification is required, please do not hesitate to let us know at any time. Thanks again for your kind help. We look forward to hearing from you soon.

Yours sincerely,

Xin Huang

School of Chemical Engineering and Technology,

Tianjin University, Tianjin 300072, P. R. China

Tel: +86-22-27405754

Fax: +86-22-27374971

Answers to the comments and suggestions raised by the reviewers and the editor:

Reviewer #4:

Point 1: Line 10-11   --->   First statement of the abstract is of little usefulness. Better start directly with “Removal of contaminants… 

Response 1: Indeed, the first statement of the abstract is a little prolix. It has been deleted as you suggested. 

Point 2: 11-16   --->   This part should be improved reducing the redundant terms (“heavy metals” and “wastewater” are mentioned 3 times each in 6 lines)

Response 2: Thanks for your suggestions. This part has been simplified as you suggested.

Point 3: 29   --->   Why biodegraded? Metals cannot be “degraded” like organic compounds. Please specify

Response 3: The word “biodegraded has been specified in the revised version, it means that heavy metals cannot be degraded by microorganisms once they are released into the environment.

Point 4: In this review, MOFs have been excluded, however they should be mentioned at least here. For example these two citations should be added: DOI: 10.1016/j.jhazmat.2012.11.011

DOI: 10.1039/C5TA04154F, Many more can be found in literature

Response 4: It’s a pity that the MOFs were not taken into consideration in this review. Thanks for your recommended references, they have been cited in the introduction to enrich the review.

Point 5: 45-47   --->   This sentence needs a reference

Response 5: The reference of this sentence has been added as you suggested.

Point 6: 52   --->   Change suggestion: “various kind of nanomaterials, including” > “the following nanomaterials:”

Response 6: Thanks for your kind suggestions for the English writing of this article, this statement has been changed as you suggested.

Point 7: 55   --->   I think this sentence should have “are” in place of “were” (present)

Response 7: Thanks for your advice. The word “were” has been changed into “are” as you suggested.

Point 8: 57-58   --->   This sentence needs a reference

Response 8: We felt sorry that the reference of this sentence was not added. The reference of this sentence has already been added as you suggested.

Point 9: 59   --->   What do you mean with “simplicity”? Please detail.

Response 9: Simplicity was not an appropriate word here, it has been substituted with “ease”, which means that it’s easy to modify the carbon-based nanomaterials.

Point 10: 68   --->   Typo: “muti” -> “multi”. In the same line CNTs are mentioned as “graphene materials”, but this is not correct. I suggest to change with “carbon-based materials”

Response 10: Thank you for pointing out my evident mistakes, they have been corrected.

Point 11: 71   --->   Please provide a range of m2/g and mg/g regarding surface area and adsorption capacity

Response 11: We felt sorry that the range of surface area and adsorption capacity could not be provided because we could not guarantee that the literatures which we learned are sufficient. Also, some literatures did not provide the surface area. So, it may be not accurate to give a range of adsorption surface. As for the adsorption capacity, CNTs-based nanomaterials could be used to remove different heavy metals, and it may be meaningless to compare their removal capacities towards different metal ions. Thus, the range of adsorption capacity of CNTs-based nanomaterials is a little meaningless to our opinions.

Point 12: 77   --->   In the caption of Figure 1, a bundle is mentioned, but this is not referred in the main text. Please fix the incongruence

Response 12: Thanks to your reminder, an explanation of Figure 1 has been given. Though the aim of this figure is to illustrate the adsorption active sites, “bundle” here is not that important.

Point 13: 80   --->   Which functional groups? Please mention some meaningful ones.

Response 13: Supplements about the functional groups have been made as you suggested.

Point 14: 84-88   --->   I do not get here where is the novelty. Please reformulate if necessary

Response 14: Thanks for your suggestion. The word “novel” here may be not appropriate, so it has been deleted to avoid misunderstandings.

Point 15: 99   --->   Is this cost issue still valid in 2019? Please check the market and correct if needed.

Response 15: Actually, the cost issue is still a problem hindering the commercial use of CNTs. But it’s a pity that some detailed figures could not be provided here.

Point 16: 114   --->   Which functional groups? Please mention some meaningful ones.

Response 16: Supplements about the functional groups have been made as you suggested.

Point 17: 116   --->   Please provide a range of m2/g regarding surface area.

Response 17: The reference we cited did not provide the surface area of graphene-based nanomaterials, and we are afraid that the range of surface area we provide is not accurate due to the limitations of our scope and knowledge.

Point 18: 117   --->   Which “ample” (?) functional groups? Please mention some meaningful ones.

Response 18: Supplements about the functional groups have been made as you suggested.

Point 19: 126   --->   Where does the humic acid come from?

Response 19: The humic acid was added into the aqueous solution. Because humic acid present widely in the nature, so the author investigated the adsorption effect with the presence of humic acid.

Point 20: 128   --->   This phrase says nothing, there should be a follow up, for example “…higher than the commonly reported adsorbents, such as...

Response 20: Thanks for your kind suggestions on the writing of this article, this phrase has been deleted from the manuscript.

Point 21: 138   --->   I think the term GOCA is not correct referred only to calcium alginate. Please check.

Response 21: Thanks for you pointing out this mistake. Indeed, COCA was a clerical error and has already been corrected.

Point 22: 146   --->   Change suggestion: “are still under the lab level” > “are still at a preliminary stage of research” or “are not commercialized yet”

Response 22: This statement has been changed as you suggested.

Point 23: 152   --->   Typo: “meal” > “metal”

Response 23: This mistake has been revised.

Point 24: 176   --->   Addition suggested: “The standard potential E0 of some environmentally relevant metals…”

Response 24: This addition has been made as you suggested.

Point 25: 234   --->   Is high temperature OR pressure, or high temperature AND pressure? Please check and add a reference.

Response 25: Thanks for your careful reading. The reference of this statement has been added.

Point 26: 241-243   --->   Please cut this in two statements.

Response 26: This statement has been cut into two parts as you suggested.

Point 27: 243   --->   Nanosized is repeated twice in the same line. Please rephrase.

Response 27: This statement has been rephrased as you suggested. Thanks for your helpful suggestions on the English writing of this article.

Point 28: 244   --->   Although for some elements the state of oxidation is trivial (Al (III), Mg(II)), for others is not. Please explicit the oxidations state of iron, manganese and cerium relevant for this review.

Response 28: The oxidations state of iron, manganese and cerium have been assigned as you suggested.

Point 29: 247-251   --->   This part is not much meaningful and it should be expanded a little more and enriched by some references, especially for the first two sentences.

Response 29: This part has been enriched as you suggested and two relevant references have been added here.

Point 30: 253   --->   Change suggestion: “often exist in the form of minerals in the nature” > “already exist as minerals in nature”. Moreover, it would be better to out “is” instead of “was” (present)

Response 30: Change has made as you suggested. Thanks for your helpful suggestions on the English writing of this article.

Point 31: 254   --->   It would be more preferable the form “thanks to” rather than “as a result of” in this part.

Response 31: Thanks for your helpful suggestions on the English writing of this article. Change has made as you suggested.

Point 32: 256   --->   I suggest to describe uranium in full rather than U to avoid confusion. Please add the uptake of uranium from the reference.

Response 32: Thanks for your helpful. Change has made as you suggested.

Point 33: 266   --->   Although Kelvin is the SI standard, please add or convert to Celsius for the readers.

Response 33: Thanks for your helpful suggestions on the English writing of this article. The temperature in the form of Celsius has been added.

Point 34: 270   --->   Please fix this phrase (“Hematite is highly resistant to corrosion and is the most stable…”)

Response 34: This phrase has been fixed as you suggested.

Point 35: 292   --->   Please provide a range of m2/g regarding surface area.

Response 35: We felt truly sorry that the surface area could not be provided because of the reason we referred to before.

Point 36: 293   --->   This sentence needs a reference.

Response 36: Because this sentence has the same reference as the sentence right after it, its reference was not addressed here.

Point 37: 300   --->   The term “regulable” does not exist. Please change with “tunable”

Response 37: This word has been substituted as you suggested. Thanks for your helpful suggestions on the English writing of this article.

Point 38: 320   --->   Change: “employ” > “use”

Response 38: This word has been substituted as you suggested. Thanks for your helpful suggestions on the English writing of this article.

Point 39: 321   --->   Change suggestion: “More importantly” > “Similarly to maghemite”, as maghemite and magnetite are structurally analogous and both have magnetic properties. In this regard, the authors should specify which and where they are talking about superparamagnetic and/or ferromagnetic systems.

Response 39: Thanks for your helpful suggestions on the English writing of this article. The change has been done as you suggested. Though the superparamagnetic and ferromagnetic systems are not discussed here because we think that it may be less important in this part. The aim of this part was to illustrate their merits of easy separation.

Point 40: 326-328   --->   Is this phrase coming from a reference or from the authors? Please motivate the claim (with bibliography).

Response 40: The phrase came from the reference of Giraldo’s work and has been given an explanation already.

Point 41: 336   --->   I suggest to put the terms “acid” and “base” as plurals.

Response 41: Thanks for your helpful suggestions on the English writing of this article. The term “acid” and “base” have been represented as plurals.

Point 42: 338   --->   Change: “protection” > “protective”

Response 42: The change has been done as you suggested. Thanks for your helpful suggestions on the English writing of this article.

Point 43: 351   --->   Please report the number of cycles, to support the reusability of the adsorbent.

Response 43: The cycle numbers have been added in this part.

Point 44: 367-376   --->   In this section, the authors should include the adsorption capacities of the mentioned systems (mg/g).

Response 44: It’s really a pity that the adsorption capacities of these materials were not mentioned and thank you for pointing out it. The adsorption capacities of the mentioned systems have been added.

Point 45: 409   --->   How small are these micropores? Please add the related values.

Response 45: The reference did not provide the data about the size of these micropores, so it’s a pity that its dimension was not mentioned in the review.

Point 46: 419   --->   It looks a bit dodgy that a Zn based system is used to uptake Zn. Please check and correct if necessary

Response 46: The article I referred to indeed reported the uptake of Zn2+ by using Zn-based nanomaterials.

Point 47: 421   --->   Please fix and report the pH range.

Response 47: The pH range has been supplemented.

Point 48: 436   --->   Please report the nanowire diameter.

Response 48: The diameter of the nanowire has been added.

Point 49: 440   --->   The term “arrestment” sounds weird. Perhaps “stop” or “termination” are more appropriate.

Response 49: The term “arrestment” has been revised as you suggested. Thanks for your helpful suggestions on the English writing of this article.

Point 50: 443-445   --->   Please rephrase this statement as it is unclear.

Response 50: This statement has been rephrased.

Point 51: 447-449   --->   Can you comment more on this sentence?

Response 51: Some enrichments on this sentence.

Point 52: 457   --->   This is too generic, the authors can focus directly on the gamma phase.

Response 52: Still, we think it’s necessary to give an introduction about the crystalline structures of Al2O3, so we felt sorry that we didn’t make changes on this sentence.

Point 53: 458   --->   Change to: “which is the most widely used”

Response 53: Change has been made as you suggested. Thanks for your helpful suggestions on the English writing of this article.

Point 54: 470-471   --->   Change suggestion to: “…have also been reported to remove Hg(II),…”

Response 54: Change has been made as you suggested. Thanks for your helpful suggestions on the English writing of this article.

Point 55: 477   --->   Bacteria names are usually written in italics.

Response 55: Change has been made as you suggested. Thanks for your helpful suggestions on the English writing of this article.

Point 56: 481, 488, 492, 496, 537, and 649   --->   Please check and correct the spacing between words.

Response 56: The spacing between words in these lines have been checked to be fine.

Point 57: 505   --->   Change to: “photocatalysis”. The following “water treatment” requires a reference.

Response 57: The change has been made as you suggested and the reference on the “water treatment” has also been added.

Point 58: 515   --->   Change suggestion: “could be finished in several minutes” > “is complete after (NUMBER) minutes”.

Response 58: The change has been made as you suggested. Thanks for your helpful suggestions on the English writing of this article.

Point 59: 521   --->   Please explain the terms SDC-F and SDC-I.

Response 59: SDC-F and SDC-I have been explained.

Point 60: 525-528   --->   This sentence needs a reference.

Response 60: The reference has been added as you suggested.

Point 61: 529   --->   This sentence needs a reference. Moreover I suggest to change “were” with “are” (present)

Response 61: The reference has been added and word “were” have been changed into “are”. Thanks for your helpful suggestions on the English writing of this article.

Point 62: 537-538   --->   Please rephrase this statement as it is unclear.

Response 62: This statement has been rephrased.

Point 63: 543   --->   Please check the terms “thiol” and “dendrimer” as they look to be placed without a rationale.

Response 63: The terms “thiol” and “dendrimer” have been revised. Thank you for pointing out this mistake.

Point 64: 546-547   --->   Unnecessary duplications should be avoid, please rephrase.

Response 64: This statement has been rephrased to avoid duplications. Thanks for your helpful suggestions on the English writing of this article.

Point 65: 550-552   --->   Here there is a list of many things but some systems looks to be the core, some systems the coating. It is unclear, please check and rephrase

Response 65: The word “coated” has been changed into “combined with” to avoid unclear meaning.

Point 66: 578   --->   Which nanocomposite(s) are the authors referring to? What is the core/inorganic part involved here?

Response 66: The nanocomposites here are referred to polymer-supported nanocomposites. Two polymers we referred to were off the point and have been deleted. As for the core/inorganic part involved here, we are sorry that we don’t think it’s necessary to be put here.

Point 67: 606   --->   The term “poisonousness” sounds weird. Perhaps “toxicity” is more appropriate.

Response 67: The word has been substituted as you suggested. Thanks for your helpful suggestions on the English writing of this article.

Point 67: 613, 614, 616, and 618   --->   What does ferric/ferric oxide mean? Maybe the authors refers to iron and iron oxide? Please fix these terms.

Response 67: The terms have been fixed as you suggested. Thanks for your helpful suggestions on the English writing of this article.

Point 68: 632   --->   The high cost of magnetic nanoparticles is fairly debatable nowadays. If so, please state some figures in regard and compare with current adsorbents.

Response 68: The figures about the cost of magnetic nanoparticles cannot be found. Since the cost of magnetic nanoparticles is debatable today, the statement on their costs has been deleted.

Point 69: 638   --->   Change “have exhibited” with “exhibit”

Response 69: Change has been made as you suggested. Thanks for your helpful suggestions on the English writing of this article.

Point 70: 641   --->   Please define this difficulty, as there are many standard processes that can be used, such as filtration, centrifugation, flocculation, which are possible also at the industrial scale.

Response 70: This difficult has been defined as you suggested.

Point 71: 646   --->   Change “need to be paid” with “are needed”.

Response 71: Change has been done as you suggested. Thanks for your helpful suggestions on the English writing of this article.

Point 72: 653-655   --->   What do the authors mean with “it is high time”? Please rephrase.

Response 72: This statement has been rephrased. Thanks for your helpful suggestions on the English writing of this article.

Point 73: 656-661   --->   The last sentence is not enough and too general, the authors should put more efforts in providing a more meaningful closure.

Response 73: The closure has been enriched a little. Thanks for your helpful suggestions on the closure of this article.

Point 74: 662   --->   A summary table listing all the uptakes mentioned in this review, with the related literature references, is helpful for the reader and is strongly recommended.

Response 74: Thanks for your kind suggestions. A summary table has been given as you suggested.

Round  2

Reviewer 1 Report

The manuscript has been strongly improved through revision and can now be accepted for publication.

Author Response

Answers to the comments and suggestions raised by the reviewers and the editor:

Reviewer #1:

Point 1: The manuscript has been strongly improved through revision and can now be accepted for publication.

Response 1: Thanks very much for all your suggestions on this manuscript.

Reviewer 3 Report

Dear Editor,

In my belief, the corrections applied in the revised version are convincing enough. However:

1- The authors need to widely compare the novelty of their work with the suggested similar review paper (my former comment 1). Only mentioning that a similar work is not comprehensive is by no means scientific and acceptable.

2- The authors must give a brief theoretical information regarding the kinetic models, isotherms etc. for the reader so that the reader comprehends what pseudo second order kinetic model means, for instance. The authors have solely added some words about the name of adsorption model!! Page 3, line 115, e.g.

3- Scheme 2, what the graphical elements imply? spheres, curved lines, shell, core etc. They should be named inside the scheme.

4- Scheme 3 is the blue frame encompassing the Figure necessary?

Author Response

Answers to the comments and suggestions raised by the reviewers and the editor:

Reviewer #3:

Point 1: The authors need to widely compare the novelty of their work with the suggested similar review paper (my former comment 1). Only mentioning that a similar work is not comprehensive is by no means scientific and acceptable.

Response 1: We are sorry that the enrichments of this part were still lacking. More detailed comparisons have been added in the revised manuscript considering your suggestions, as highlighted by red. Thanks very much for your suggestions.

Point 2: The authors must give a brief theoretical information regarding the kinetic models, isotherms etc. for the reader so that the reader comprehends what pseudo second order kinetic model means, for instance. The authors have solely added some words about the name of adsorption model!! Page 3, line 115, e.g.

Response 2: We are sorry that we did not get your meaning for the first revision. Detailed introduction about the kinetic models are added into the revised manuscript as “2. Adsorption isotherms and kinetics”. Indeed, the information regarding the adsorption isotherms and kinetics is vital for the illustration of adsorption behaviours. Thanks for your helpful suggestions.

Point 3: Scheme 2, what the graphical elements imply? spheres, curved lines, shell, core etc. They should be named inside the scheme.

Response 3: Thanks for your helpful suggestions. Explanations about different elements have been added into Scheme 2 in the revised manuscript. We hope this revision can be to your satisfactory.

Point 4: Scheme 3 is the blue frame encompassing the Figure necessary?

Response 4: The blue frame in the Scheme 3 has been deleted as you suggested. Thanks for your helpful suggestion.
